# Commensal colonization of *Candida albicans* in the mouse gastrointestinal tract is mediated via expression of candidalysin and adhesins

Kelsey E. Mauk,[1] Pedro Miramón,[2] Michael C. Lorenz,[2] Léa Lortal,[3] Julian R. Naglik,[3] Bernhard Hube,[4,5,6] Lynn Bimler,[1] Farrah Kheradmand,[1,7,8] David B. Corry[1,7,8,9]

**ABSTRACT**  The ubiquitous fungal pathogen *Candida albicans* has the potential to either asymptomatically colonize the gastrointestinal (GI) tract or become an invasive pathogen through mechanisms that remain incompletely understood. Here we explored the fungal, host, and environmental factors that influence the ability of *C. albicans* to colonize the mouse GI tract using a representative clinical strain, CLCA10. After a single gavage challenge ($5 \times 10^6$ CFU *C. albicans*), specific pathogen-free (SPF) mice remained colonized with *C. albicans* strain CLCA10, but not other *Candida* species, for at least 58 days with the fungus confined largely to the gut luminal contents. Colonized mice exhibited no weight loss or other signs of active infection, and CLCA10 did not disrupt the gut microbiome. Moreover, *C. albicans* colonization with CLCA10 was not substantially affected by the mouse commercial source or the method used to cultivate the fungus prior to gavage. Although some genetically manipulated *C. albicans* strains were unable to robustly colonize, strain SC5314 also colonized the mouse gut despite having enhanced pathogenicity. *C. albicans* CLCA10 gut colonization in part depended on the hypha-associated adhesins Als3 and Hwp1 and the peptide toxin candidalysin and could not be eradicated by potent antifungal therapy. Thus, this study concludes that *C. albicans* gut colonization in the mouse is critically dependent on fungal hyphal factors, the targeting of which could enhance strategies to reduce *C. albicans* gut colonization and the intractable threat to human health it represents.

**IMPORTANCE** *Candida albicans* is an important human fungal pathogen and a ubiquitous colonizer of the gastrointestinal (GI) tract. However, it is not understood how *C. albicans* persists within the GI tract and from which it may disperse to cause disease. Here, we demonstrated that multiple strains of *C. albicans*, including the widely used SC5314 strain, robustly colonize the mouse GI tract for at least 2 months. This colonization caused no disruption to the host tissue or bacterial microbiome and was resistant to clearance by antifungal drugs. Importantly, colonization was mediated by proteins expressed by *C. albicans* that are known to be involved in fungal virulence and unrelated to experimental conditions. Overall, this work identifies mechanisms by which *C. albicans* persists in the GI tract, enhancing our knowledge of host-fungal interactions during commensal colonization and potentially how to reduce such colonization.

**KEYWORDS**  *Candida albicans*, intestinal colonization, virulence factors

*Candida albicans* is an opportunistic pathogen that colonizes and infects multiple mucosal surfaces of humans, including the oral cavity, urogenital tract, upper and lower airway, skin, and gastrointestinal (GI) tract (1). *C. albicans* exhibits broadly pathogenic potential, producing disease in many organs, and can cause both superficial

Address correspondence to David B. Corry, dcorry@bcm.edu, or Lynn Bimler, lynn.bimler@bcm.edu.

The authors declare no conflict of interest.

See the funding table on p. 20.

mucosal infections and severe, invasive life-threatening systemic disease (2, 3). In addition to these infectious complications, *C. albicans* has also been associated with multiple chronic illnesses, including inflammatory bowel disease (4–6), allergic airway disease (7–9), alcohol-associated liver disease (10), and chronic brain disease (11–13). Accordingly, infections with *C. albicans* stimulate robust immunity both at mucosal sites and systemically (14–16). However, despite the pathogenic potential of *C. albicans* and the ability of healthy hosts to effectively respond to *C. albicans* infection, *C. albicans* resides commensally in the GI tract of over 80% of healthy individuals (17, 18).

Gut colonization with *C. albicans* confers protective immunological benefits for the host, including the expansion of fungal-reactive Th17 cells, which can protect against invasive infections due to *C. albicans* and likely other fungi and non-fungal pathogens (14, 19). However, *C. albicans* gut residence also imparts an increased risk of serious systemic and inflammatory disease (20). Many systemic *C. albicans* infections are initiated by gut-resident strains (21–23), suggesting that systemic infections are primarily caused by translocation of *C. albicans* cells from the GI tract. Furthermore, the accumulation of pathological anti-*C. albicans* adaptive immune cells generated during gut colonization is observed in both intestinal and pulmonary inflammatory diseases (9). Finally, the intestinal presence of *C. albicans* can be correlated with increased disease severity in ulcerative colitis (UC) (24, 25), and hypervirulent strains of *C. albicans* have been shown to aggravate UC-associated intestinal inflammation and damage (6). Together, these observations suggest that both the behavior of *C. albicans* in the GI tract and the systemic host immune response to the fungus during colonization may be detrimental to human health. Therefore, to understand how *C. albicans* both initiates systemic infection and contributes to chronic disease, it is imperative to understand how this ubiquitous fungus establishes its commensal lifestyle and perpetuates long-term gut residency.

The impact of *C. albicans* colonization of the GI tract and other mucosal barriers and the subsequent host immune responses has been widely studied using mouse models. However, as *C. albicans* is not universally considered a natural colonizer of the GI tract of commonly used mouse strains (26), most *C. albicans* colonization studies have utilized antibiotics (27–30) or germ-free animals (30–32) to attenuate or remove members of the gut microbiome normally causing colonization resistance and inhibiting fungal engraftment, or immune suppression (33–35) to inhibit fungal clearance. These studies have yielded useful insights into the biological behavior of *C. albicans* in the GI tract but did not accurately delineate how *C. albicans* colonizes and interacts with a healthy host under normal physiological conditions, including a balanced microbiome.

*C. albicans* interacts with many bacterial species, including antagonistic and synergistic relationships (36). While some bacteria antagonize *C. albicans* through multiple mechanisms, others influence its virulence and allow *C. albicans*, for example, to form highly pathogenic biofilms (37–40). Depletion of the bacterial microbiome with broad-spectrum antibiotics likely disrupts these interkingdom interactions, thereby changing fungal behavior and altering interactions within the host GI tract. Moreover, immunosuppression may facilitate the establishment of a GI niche, but may also inhibit fungal behaviors normally deployed to evade immune-mediated destruction. For example, IgA present in the GI tract substantially changes *C. albicans* behavior with regard to gut colonization and virulence (41, 42). Similarly, gut-resident phagocytes change the diversity and composition of the intestinal mycobiome and protect against fungal-exacerbated intestinal disease (43). Thus, interventions that alter host immunity and the microbiome obscure the natural behavior of *C. albicans* and related species, precluding our understanding of how *C. albicans* persists in the native GI tract and potentially contributes to diverse diseases.

Using a clinical (respiratory tract) isolate (CLCA10), we have developed a simple, long-term model of *C. albicans* colonization of the unmanipulated murine GI tract that offers the opportunity to observe more authentic interactions between *C. albicans* and a native GI tract microbiome. After a single gavage challenge, specific pathogen-free (SPF) mice remained colonized with the CLCA10 strain of *C. albicans,* but not other

*Candida* species, for at least 58 days with the fungus almost entirely confined to the gut luminal contents. Colonized mice exhibited no weight loss or other signs of active infection. Moreover, *C. albicans* colonization was not substantially affected by the mouse commercial source or by the method used to cultivate the fungus prior to gut colonization and in part depended on the hyphal adhesins Als3 and Hwp1 and the peptide toxin candidalysin. Potent antifungal therapy was unable to clear *C. albicans* from the mouse GI tract. Although some genetically manipulated *C. albicans* strains were unable to robustly colonize the murine GI tract, the widely used and highly pathogenic SC5314 strain also exhibited long-term gut colonization. Under these conditions, where the unmanipulated host was colonized with the relatively non-pathogenic CLCA10 strain of *C. albicans*, fungal colonization did not substantially disrupt the gut microbiome. Thus, multiple *C. albicans* strains were well-adapted to silently persist in the mouse GI tract. The adaptations allowing *C. albicans* gut persistence likely facilitate the ubiquitous spread of *C. albicans* throughout human and other mammalian populations and underscore the threat such colonization represents under conditions favoring the transition from colonizing organism to invasive pathogen.

## RESULTS

### *C. albicans* colonizes the murine GI tract for a minimum of 58 days

Distinct *C. albicans* strains have been shown to colonize the mouse GI tract, although SC5314, the most commonly used reference strain of *C. albicans*, is thought to be a poor colonizer of the GI tract (26) in part because (i) SC5314 more readily forms damaging hyphae that penetrate epithelial cells during mucosal contact and thus inhibit its ability to establish a commensal GI tract niche (26, 44) and (ii) it is more susceptible to antimicrobial peptides (26). SC5314 was isolated from a patient with disseminated disease and possesses a rare mutation in the ROB1 gene that in part explains its enhanced pathogenicity that is marked by increased expression of fungal ECE1 (the gene encoding for the peptide toxin candidalysin), increased host cell damage (quantified by the release of lactate dehydrogenase, LDH), and increased ERK-MAPK signaling (via EGFR, MKP1, and c-Fos) in epithelial cells and which distinguishes it from most other *C. albicans* strains (45) (Fig. S1A through S1C). As compared to SC5314, *C. albicans* strain CLCA10 that was isolated from the respiratory tract of an asthma patient displayed significantly reduced virulence indices (Fig. S1A through S1C), behaving similarly to 529L, another *C. albicans* strain that exhibits reduced virulence as compared to SC5314 due to reduced secretion of candidalysin (46). Together, these findings show that CLCA10 and SC5314 are functionally distinct *C. albicans* strains that exhibit contrasting pathogenicity in an *in vitro* system.

To assess whether CLCA10 can colonize and persist within the mouse GI tract in the absence of antibiotic use, immunosuppression, or germ-free conditions, we orally administered either PBS or $5 \times 10^6$ CFU *C. albicans* once to SPF C57BL/6 mice. We then assessed the degree of colonization in these mice by collecting and culturing the small intestinal (SI) luminal contents and homogenized intestinal tissue (including the intestinal mucus layer) at distinct timepoints over the next 2 months. We found that *C. albicans* rapidly equilibrated to colonize and stably persist within the mouse GI tract, as the fungus was consistently and stably cultured from both the SI luminal contents and homogenized intestinal tissue of mice for 2 months after oral gavage (Fig. 1A and B). Although significantly more *C. albicans* cells can be cultured from intestinal samples if plated without a freeze-thaw cycle, samples were frozen prior to being cultured due to the inability to immediately process all freshly collected tissue samples. Therefore, reported fungal burdens underrepresent the actual fungal burdens by approximately 10%–40% (Fig. S2). Neither *C. albicans* nor other culturable fungi were isolated from the SI contents or tissue of mice only given PBS (Fig. 1A and B).

To determine whether *C. albicans* CLCA10 colonization leads to intestinal pathology in the host, we assessed H&E-stained tissue sections from the small and large intestines for epithelial damage and immune cell infiltration and recorded weight change over time.

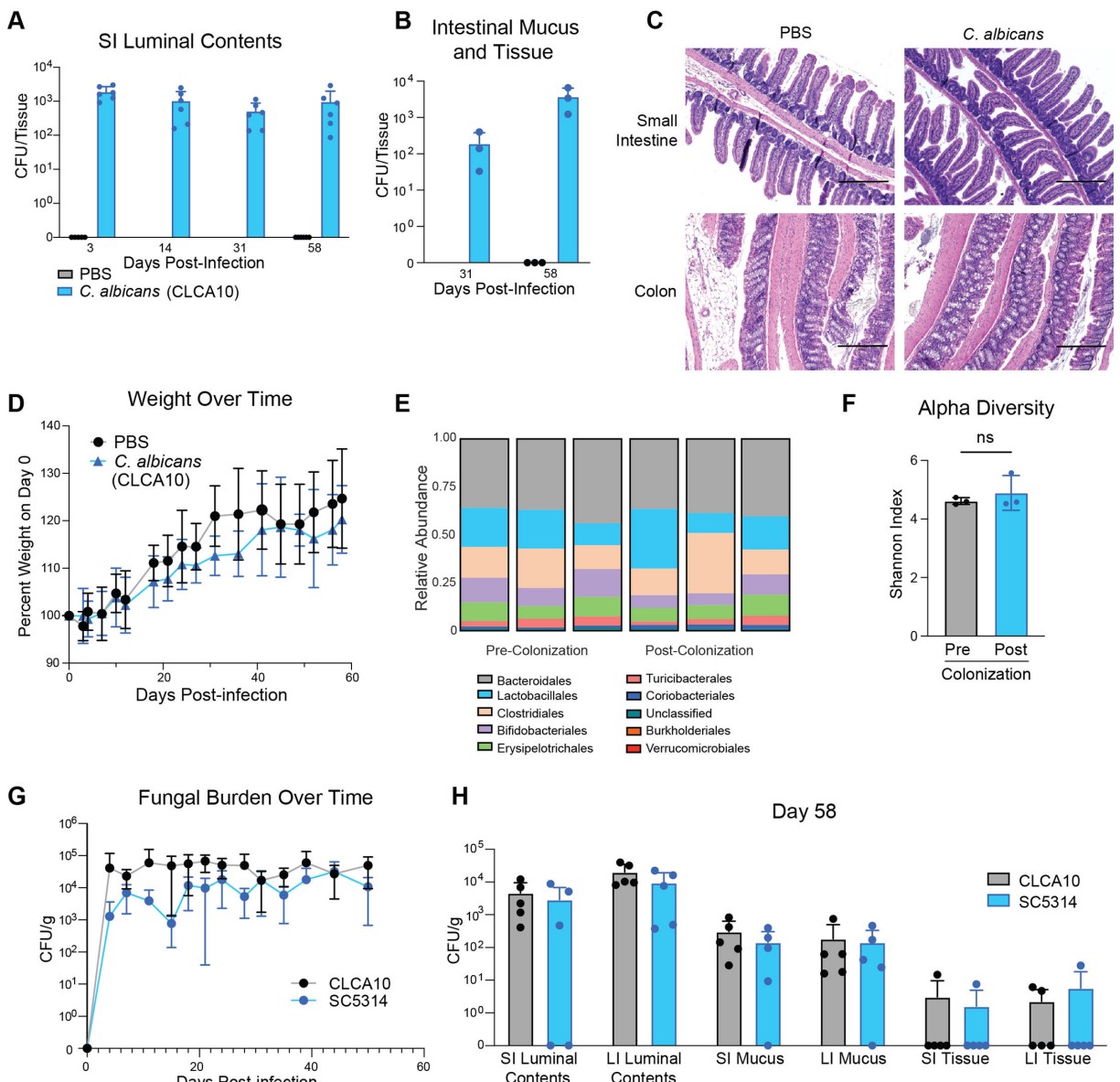

**FIG 1** Mice remain colonized 58 days after oral infection with *C. albicans*. (A–D) Mice were given either PBS or *C. albicans* (CLCA10) via oral gavage. CFU *C. albicans* cultured from (A) SI luminal contents and (B) homogenized SI and LI intestinal tissue of mice infected with *C. albicans* (CLCA10) or given PBS. (**C**) Hematoxylin and eosin staining of SI and colonic tissues from three days post-infection. Scale bar = 300 µm. (D) Percent of weight of mice on day 0. (E) Relative abundance of bacterial orders and (F) Shannon index of the murine fecal microbiota pre- (collected immediately prior to infection) and post-colonization (collected 14 days post-infection) with *C. albicans* (CLCA10). (G–H) Mice were infected via oral gavage with *C. albicans* (CLCA10 or SC5314). CFU/g *C. albicans* cultured from (G) fecal pellets collected from mice over time or (H) luminal contents, mucus, or homogenized tissue of the SI and LI of mice on day 58. Graphs display mean ± SD from 3 to 6 mice per group. Significance determined using Student's t-test. ns = not significant. Data representative of two experiments yielding comparable results.

We found no differences in intestinal pathology (Fig. 1C) or weight change (Fig. 1D) between colonized and uncolonized mice, suggesting that *C. albicans* colonization does not cause significant intestinal damage or inflammation under the gavage challenge conditions used.

## Intestinal colonization with *C. albicans* does not alter the bacterial microbiome

In mice in which the intestinal mucosa and microbiome have been partially disrupted by chemotherapy, introduction of the highly pathogenic *C. albicans* strain SC5314 leads to further dysbiosis, especially loss of bacterial diversity (47). However, others have shown that intestinal colonization with *C. albicans* SC5314 has a limited effect on the bacterial microbiome (26). To determine whether intestinal colonization with more typical, hypovirulent, and commensal-like *C. albicans* leads to significant changes in the composition of the bacterial microbiota under steady state conditions, we characterized the bacterial microbiome via 16S sequencing of mouse fecal pellets collected pre- and post-gavage challenge with *C. albicans* strain CLCA10 ($5 \times 10^6$ yeast cells). Consistent with prior results using SC5314 (26), we found that colonization with *C. albicans* CLCA10 did not significantly affect the composition of the bacterial microbiota at the order level, which remained comparable before and after *C. albicans* colonization (Fig. 1E). Furthermore, we found no significant difference in alpha diversity (measured via Shannon index) upon colonization with *C. albicans* (Fig. 1F), suggesting that under the specific conditions used, introduction of commensal-like *C. albicans* to the mouse GI tract does not initiate broad shifts in the composition of the bacterial microbiome.

## Multiple strains of *C. albicans* can persist in all compartments of the GI tract

To establish whether the ability to colonize the mouse GI tract is a strain-specific quality and to further investigate which intestinal compartments are enriched for *C. albicans*, we gavage-challenged mice with either CLCA10 or SC5314 and assessed fungal burdens of both strains from fecal pellets, luminal contents, mucus, and tissue of the small and large intestines after 2 months. Interestingly, we found that both fungal strains persisted in the GI tracts of all mice for a minimum of 2 months, with no significant difference in overall fecal burdens that ranged from $10^3$ to $10^5$ CFU/gram (Fig. 1G). Furthermore, upon culturing of different intestinal compartments, we found that both strains of *C. albicans* were present in all compartments of the intestine, including the intestinal mucus and tissue, with no significant differences in fungal burden between groups (Fig. 1H). Because *C. albicans* intestinal burdens remained relatively unchanged throughout the experiment (Fig. 1G), we used 14 days post-colonization as our experimental endpoint for subsequent experiments to account for any immediate impact of the host immune response. Furthermore, although both strains effectively colonize the mouse GI tract, we chose to conduct subsequent experiments using CLCA10 as it is more representative of a commensal, non-hypervirulent strain of *C. albicans* found in the general community.

## Mouse strain but not sex alters *C. albicans* colonization potential

We next determined whether host factors alter the ability of *C. albicans* CLCA10 to persist in the mouse GI tract. We found that the degree of colonization was not dependent on biological sex, as both male and female mice remained colonized to a similar degree (Fig. 2A). Although mice from different commercial sources have distinct intestinal microbiomes (48), *C. albicans* CLCA10 readily colonized age- and sex-matched mice purchased from different vendors (Fig. 2B).

We further determined whether *C. albicans* colonization is affected by the strain of the mouse host, as different laboratory mouse strains exhibit different immune biases due to genetic drift (49), which may affect the ability of the host to clear fungi. We found that BALB/c and 129S1/SvImJ mice exhibited significantly reduced fungal recovery from the large intestinal (LI) contents compared to C57BL/6 mice, suggesting that these strains are more resistant to *C. albicans* colonization of the GI tract (Fig. 2C). Regardless of these differences, all mice from all strains of mice remained colonized 14 days after infection, demonstrating that *C. albicans* CLCA10 can persist within the wild type, unmanipulated mouse GI tract regardless of genetic background and the mucosal immune response generated after 2 weeks of colonization.

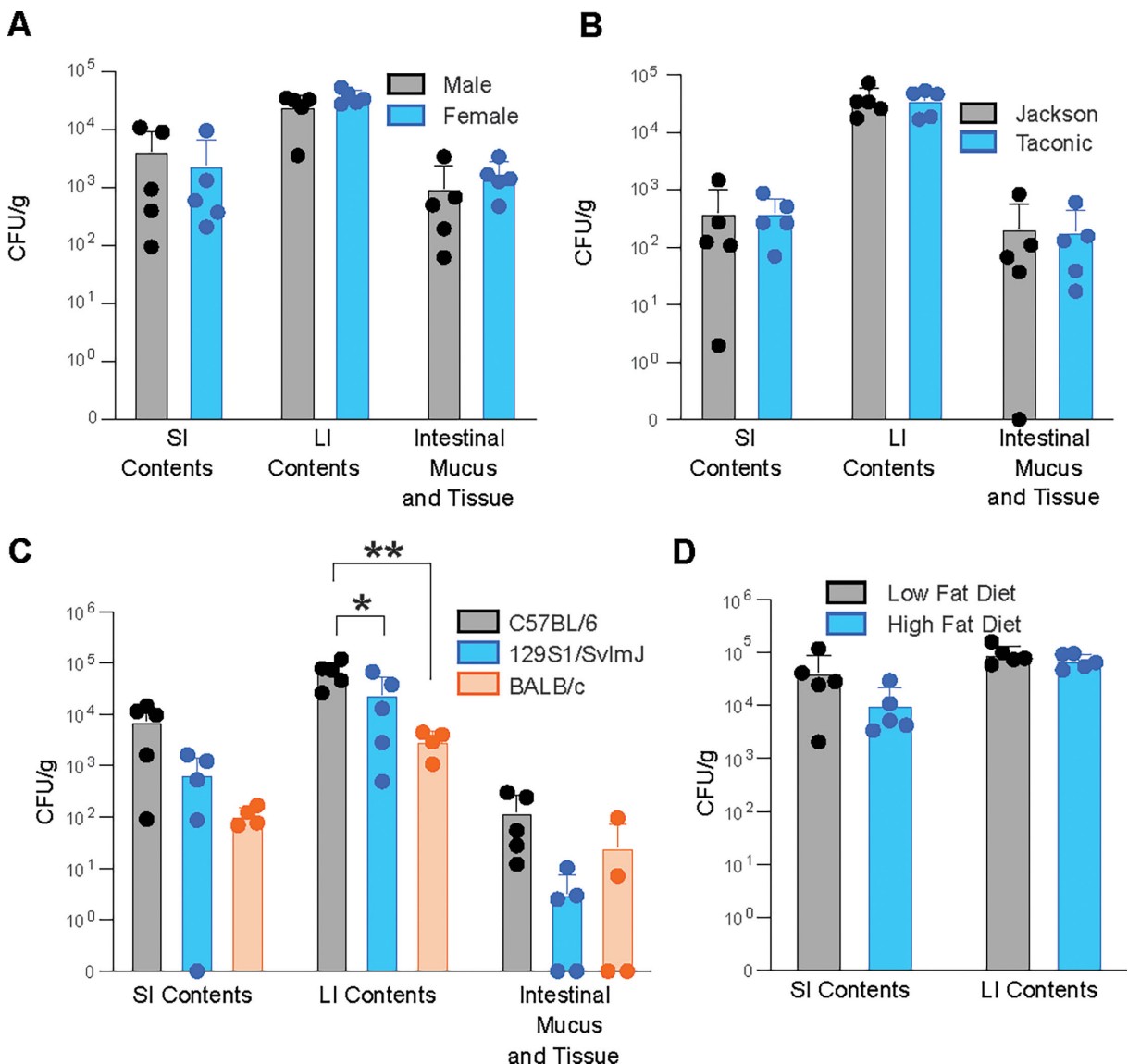

**FIG 2** Colonization is not dependent on mouse sex, vendor, strain, or consumption of a high-fat diet. CFU/g *C. albicans* (CLCA10) cultured from SI contents, LI contents, and homogenized intestinal mucus and tissue of (A) male and female C57BL/6 mice, (B) female C57BL/6 mice from Jackson Labs or Taconic Biosciences, (C) female C57BL/6, BALB/c, and 129S1/SvImJ mice, and (D) female C57BL/6 mice fed either a low-fat or high-fat purified diet infected via oral gavage with *C. albicans*. Samples were collected 14 days post-infection. Graphs show mean ± SD from 4 to 5 mice per group. Significance determined using (A, B, and D) Student's t-test with Holm-Šídák correction or (C) one-way ANOVA with Dunnett's test. *$P < .05$, **$P < .01$. Data representative of two experiments yielding comparable results.

## High-fat diet does not change *C. albicans* colonization

In addition to factors inherent to the host, we sought to determine whether diet, another host-associated factor, could affect the ability of *C. albicans* CLCA10 to colonize the GI tract. Consumption of a high-fat diet alters intestinal immunity and permeability (49, 50) and the content and diversity of the gut microbiome (51). Furthermore, the lipid content of rodent diets influences the degree of colonization of *C. albicans* in complex ways that depend in part on the dietary lipid composition (52, 53). To further determine whether dietary fat content influences the colonization potential of *C. albicans*, mice were fed a purified diet containing either 10% or 60% of calories derived from fat that is distinct from the diet used in previous experiments, for 5 days before *C. albicans* oral gavage and were maintained on the same diet for the duration of the experiment. We found that

consumption of a unique purified diet resulted in no significant difference in the degree of *C. albicans* CLCA10 colonization between mice fed low-fat or high-fat diets (Fig. 2D).

## *C. albicans* culture conditions before gavage do not change colonization potential

In addition to host factors, we determined how differences in preparation of the *C. albicans* CLCA10 inoculum prior to gavage influenced colonization potential. *C. albicans* grown under different conditions affects its morphology and gene expression profile (54, 55). Therefore, we examined whether *C. albicans* growth media, culturing time, and temperature before gavage can alter its fitness to colonize the GI tract (Fig. 3A). We found that, although significantly fewer colonies were recovered from the LI contents of mice infected with plate-grown yeast, growth medium did not alter the ability of *C. albicans* to colonize, as yeast grown in YPD broth and on Sabouraud's agar were both able to persist out to 14 days post-gavage (Fig. 3B). Notably, *C. albicans* also effectively colonized the mouse GI tract immediately after thawing, without *in vitro* culturing (Fig. 3B). Furthermore, we found that growth temperature did not significantly affect the ability of *C. albicans* to colonize the mouse GI tract, since yeast grown at 37°C and room temperature (approximately 22°C) both colonized the GI tract to a similar degree (Fig. 3C). These experiments were replicated with the *C. albicans* SC5314 strain, and no significant differences were found between all conditions (Fig. 3D and E), although generally fewer SC5314 CFU were recovered across the different conditions tested, as expected.

## Administration of amphotericin B reduces *C. albicans* burden in the GI tract, but does not result in fungal clearance

After determining that *C. albicans* persists within the mouse GI tract under many different conditions, we investigated whether *C. albicans* colonization could be disrupted or eradicated with the administration of antifungals. Amphotericin B is a broad-spectrum fungicidal antifungal that is nearly identical to nystatin, both of which interact with fungal ergosterol to disrupt cell membranes (56–58). Although amphotericin B is highly effective in treating systemic *C. albicans* infections (59), it is unclear whether it can be used to target *C. albicans* within the intestine. We hypothesized that administration of amphotericin B to mice would decrease the degree of colonization by *C. albicans*. To test this, mice colonized with *C. albicans* CLCA10 were given either amphotericin B liposomes (AmBisome, Gilead Sciences) or vehicle via intraperitoneal injection and orally via MediGel Sucralose for 14 days (Fig. 4A). Monitoring of the intestinal fungal burden via fecal pellet culturing showed a trend in decreasing fungal recovery from the antifungal-treated group (Fig. 4B). Similarly, although amphotericin B treatment resulted in a trend in the reduction of the overall intestinal burdens of *C. albicans* in multiple intestinal compartments, *C. albicans* was not completely cleared from any of the mice (Fig. 4C). These studies suggest that standard treatment regimens of the highly effective antifungal Amphotericin B are unlikely to completely clear *C. albicans* from the mouse GI tract.

## Non-*C. albicans* yeast species do not persist in the mouse GI tract

Other *Candida* species commonly inhabit the human GI tract, including *C. auris, C. tropicalis,* and *C. glabrata* (*Nakaseomyces glabratus*) as previously described (23, 60, 61). To determine whether these species can colonize and persist in the mouse GI tract as readily as *C. albicans* in our colonization model, we gavaged mice with each of these fungi. Compared to mice colonized with *C. albicans*, we recovered significantly fewer CFU from the intestinal contents and tissue of mice colonized with the non-*C. albicans* species, with most mice clearing the yeast completely after 14 days (Fig. 5A).

We also determined whether *Saccharomyces cerevisiae*, a pathologically inert yeast found in many food products and commonly isolated from the GI tract of humans (17, 61), can persist in the mouse GI tract. We found that *S. cerevisiae* was unable to persist

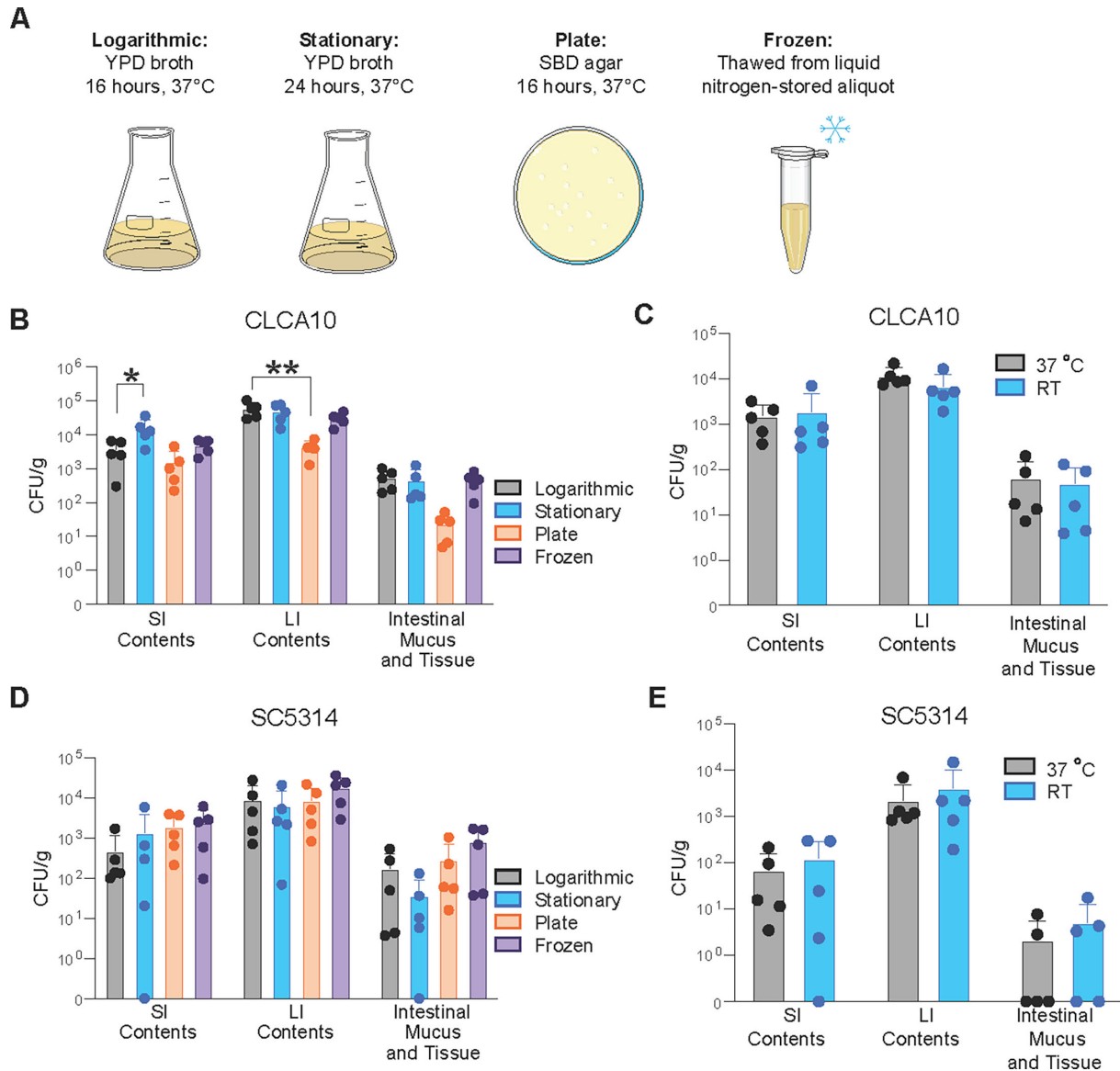

**FIG 3** Colonization is independent of inoculum preparation. (**A**) Culture conditions used in **B** and **D**. (**B–E**) CFU/g *C. albicans* cultured from SI contents, LI contents, and homogenized intestinal mucus and tissue of mice infected with (B and C) CLCA10 or (D and E) SC5314 *C. albicans*. Samples were collected 14 days post-infection. Graphs show mean ± SD from 5 mice per group. Significance determined using (B and D) one-way ANOVA with Dunnett's test or (C and E) Student's t-test with Holm-Šídák correction. *$P < .05$, **$P < .01$. Data representative of two experiments yielding comparable results.

within the GI tract, as no yeast was grown from any intestinal compartment of mice gavaged with *S. cerevisiae* after 14 days (Fig. 5B). Together, these data suggest that robust and persistent colonization of the mouse GI tract is an attribute that is not shared by all human-associated fungal species.

## Virulence factor-deficient strains of *C. albicans* exhibit reduced colonization potential

*C. albicans* expresses many proteins that directly contribute to its virulence both on mucosal surfaces and systemically, including adhesins (such as agglutinin-like sequence 3, encoded by ALS3, and hyphal wall protein 1, encoded by HWP1), proteases (including secreted aspartyl protease 2, encoded by SAP2), and a secreted peptide toxin (candidaly-sin, encoded by ECE1). These factors are critical for processes such as breaching epithelial barriers and lysing host cells to acquire additional nutrients and escape elimination by

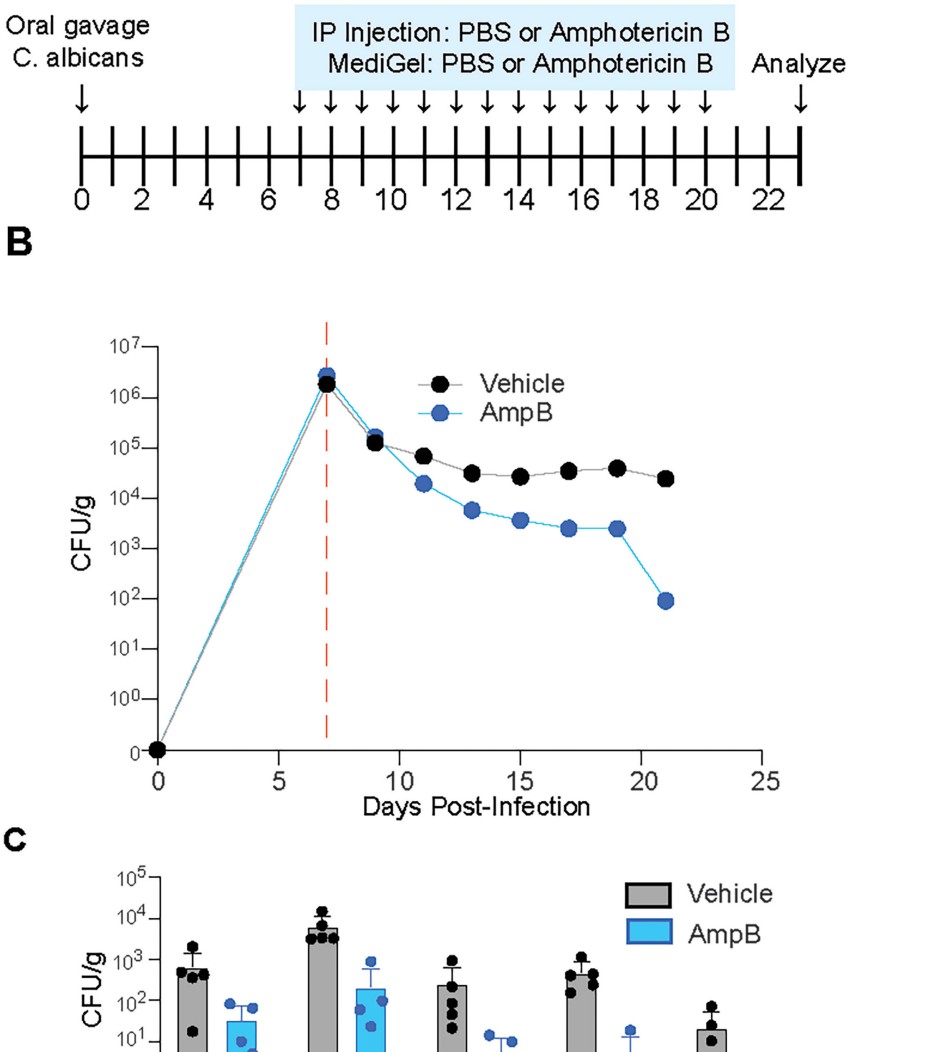

**FIG 4** GI colonization with *C. albicans* cannot be cleared by oral or intraperitoneal administration of Amphotericin B. (**A**) Amphotericin B treatment schedule. (**B–C**) CFU/g *C. albicans* (CLCA10) cultured from (B) fecal pellets collected from mice over time or (C) luminal contents, mucus, or homogenized tissue of the SI and LI of mice infected with *C. albicans* and treated with either PBS or AmBisome via oral delivery (MediGel Sucralose) and intraperitoneal injection daily. The red dashed line represents the initiation of antifungal administration. Graphs show mean ± SD from 5 mice per group. Significance determined using Student's t tests with Holm-Šídák correction (no significance). Data representative of two experiments yielding comparable results.

intracellular host defenses (62–67). In addition, some proteins involved in key metabolic pathways (such as isocitrate lyase, a key enzyme in the glyoxylate cycle encoded by ICL1), oxidative stress responses (such as catalase, encoded by CAT1), and endosomal trafficking (such as vacuolar protein sorting complex 51, encoded by VPS51) are also important for resistance to phagocyte killing and systemic virulence of *C. albicans* (68–70). However, it is unclear whether these proteins contribute to the colonization of the intestine. Since

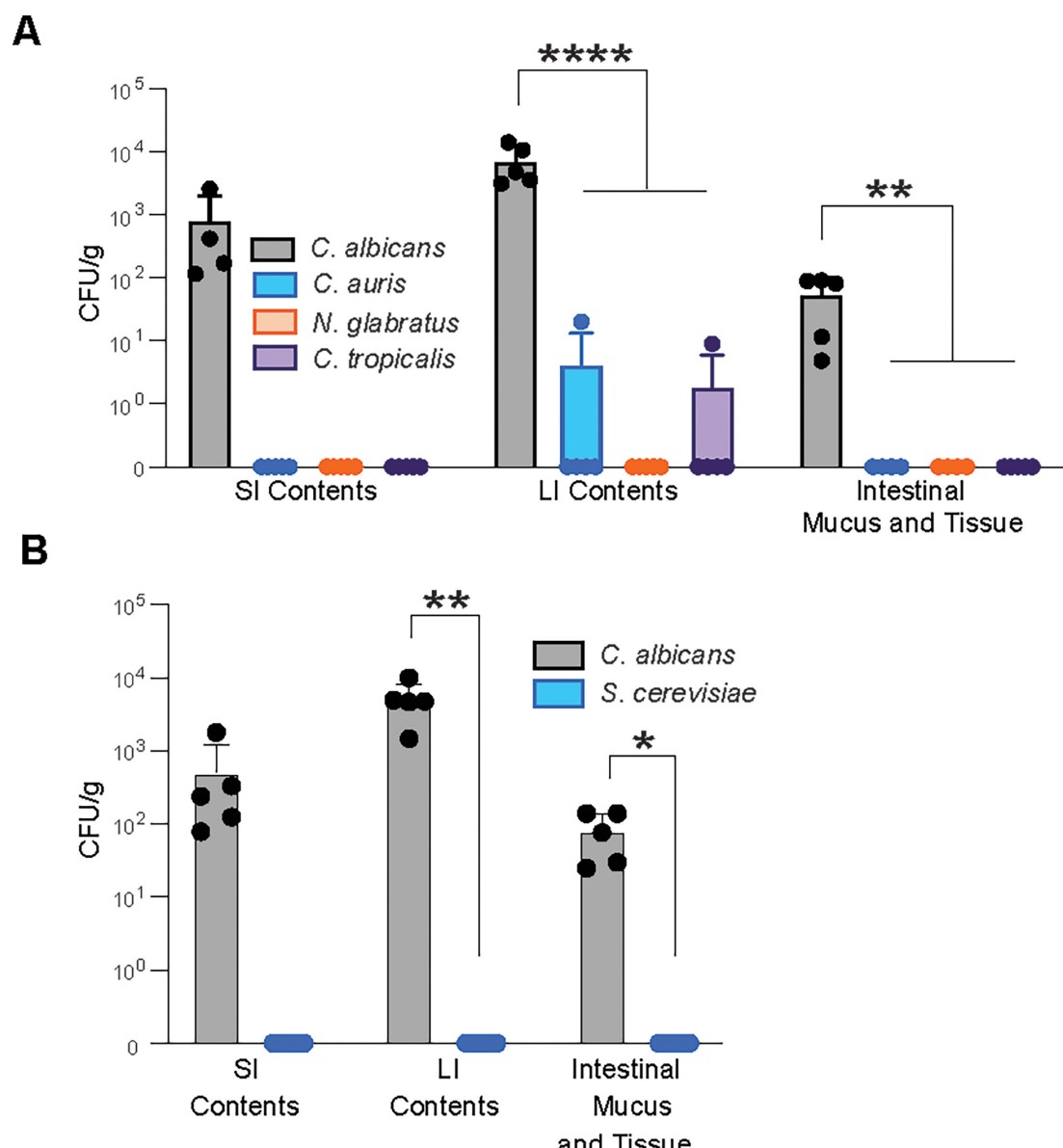

**FIG 5** Other *Candida* and non-*Candida* yeast species do not colonize the murine GI tract. CFU/g yeast grown from the SI contents, LI contents, and homogenized intestinal tissue of (A) mice infected with different *Candida* species (*C. albicans*, *C. auris*, and *C. tropicalis*) and *N. glabratus* and (B) mice infected with either *C. albicans* (CLCA10) or *S. cerevisiae*. Samples were collected 14 days post-infection. Graphs show mean ± SD from 5 mice per group. Significance determined using (A) one-way ANOVA with Dunnett's test for multiple comparisons or (B) Student's t-test with Holm-Šídák correction. *$P < .05$, **$P < .01$, ****$P < .0001$. Data representative of 1 (B) or 2 (A) experiments yielding comparable results.

*C. albicans* is not only the most virulent Candida species, but also the most common colonizer of the human gut, we hypothesized that one or more of these identified fitness or virulence factors may also facilitate persistence of *C. albicans* within the GI tract.

To determine whether fitness or virulence factors are required for intestinal colonization, we tested whether *C. albicans* mutant strains lacking these proteins retained the ability to colonize the mouse GI tract. While we found that the wild-type SC5314 strain was able to persist in the GI tract (Fig. 1H), two other SC5314-derived strains (BWP17 +Clp30 and CAI4 +Clp10), which are genetically modified variants of the parental strain SC5314 engineered to express orotidine-5′-phosphate decarboxylase, the final step in *de novo* pyrimidine biosynthesis, via the URA3 gene, exhibited severe colonization deficiency. Significantly fewer colonies were recovered from the LI contents of mice

infected with these strains (Fig. S3), and multiple mice cleared the fungus from both the luminal contents and intestinal tissue by 14 days post-infection. Thus, the genetically modified *C. albicans* strains BWP17 +Clp30 and CAI4 +Clp10 show significantly reduced potential for gut colonization as compared to an unmanipulated WT strain.

Because extant *C. albicans* mutants having deletions of virulence factors of interest are derived from the BWP17 +Clp30 and CAI4 +Clp10, we used CRISPR-Cas9 genome editing to generate *C. albicans* mutants (including *sap2△/△, als3△/△, ece1△/△, cat1△/△, icl1△/△, hwp1△/△,* and *vps51△/△*) based on the CLCA10 strain to ensure fidelity in the resulting data. We found that significantly fewer colonies were recovered from the LI contents of mice infected with the *als3△/△* and *ece1△/△* strains of *C. albicans* compared to mice infected with CLCA10, with the candidalysin-deficient strain being completely cleared from all compartments 14 days after infection (Fig. 6A). However, we found no significant reduction in the colonies recovered from mice infected with the *sap2△/△, cat1△/△, icl1△/△,* and *hwp1△/△* strains (Fig. 6A and B). While we found that the *vps51△/△* strain was incapable of persisting in the murine GI tract (Fig. 6B), this strain exhibited poor growth *in vitro*, with the 250,000 CFU inoculum taking 36 hours to grow instead of 16 hours like other strains. This suggests that the inability of the *vps51△/△* strain to colonize may be due to its generally limited growth potential rather than a specific deficiency in intestinal colonization potential.

Our observation that the expression of candidalysin is necessary for *C. albicans* CLCA10 to colonize mice with an intact microbiome reflects the findings of Liang et al. (71), who found that multiple wild-type *C. albicans* outcompetes *ece1△/△* in conventional, but not antibiotic-treated or germ-free mice. Indeed, we found that the addition of antibiotics in our model rescued the ability of the *ece1△/△* strain to colonize (Fig. 6C), suggesting that candidalysin facilitates colonization of the GI tract via interactions with the host microbiome.

To further investigate the importance of adhesins for intestinal colonization, particularly for adherence to intestinal mucus, we compared the fungal burden of mice infected with WT *C. albicans* CLCA10 with those infected with the *als3△/△* or *hwp1△/△* strains over an extended period and separated intestinal mucus from tissue at the time of tissue harvest. We found a significant reduction in *C. albicans* cells cultured from fecal pellets at two timepoints from the mice infected with the *als3△/△* and *hwp1△/△* strains, with trending decreases at almost every timepoint (Fig. 6D). In addition, we found that significantly fewer colonies were recovered from the SI contents of the mice infected with the *als3△/△* and *hwp1△/△* strains, and from the LI mucus of the mice infected with the *hwp1△/△* strain, and trending decreases in almost all other tissue compartments (Fig. 6E). These data indicate that while Sap2, Cat1, and Icl1 are dispensable for mouse GI tract colonization, Als3 and Hwp1 subtly influence the degree of GI tract colonization, while candidalysin is critical for persistence in the GI tract in the context of an intact microbiota.

## DISCUSSION

*C. albicans* is one of the most widely studied human-associated fungi, largely due to its near-ubiquitous prevalence and ability to cause serious disease among various organ systems. Moreover, anti-Candida immunity is widespread in human populations (9, 72), and in addition to potent anti-Candida innate immune responses (43, 73, 74), *C. albicans* is the main driver of antifungal Th17 responses (9). Despite this, *C. albicans* is paradoxically able to cause disease throughout the human lifespan, including recrudescent disease such as vulvovaginal candidiasis (75). The mechanisms that govern the commensal-to-pathogen shift remain unclear.

Few studies have investigated the mechanisms by which *C. albicans* grows as a commensal in the intestines of mouse strains under normal physiologic conditions, as *C. albicans* is not considered a natural colonizer of the murine GI tract. In this study, we have demonstrated that multiple strains of *C. albicans*, including the commonly used SC5314 strain, can colonize and persist within the GI tract of mice without the addition of factors

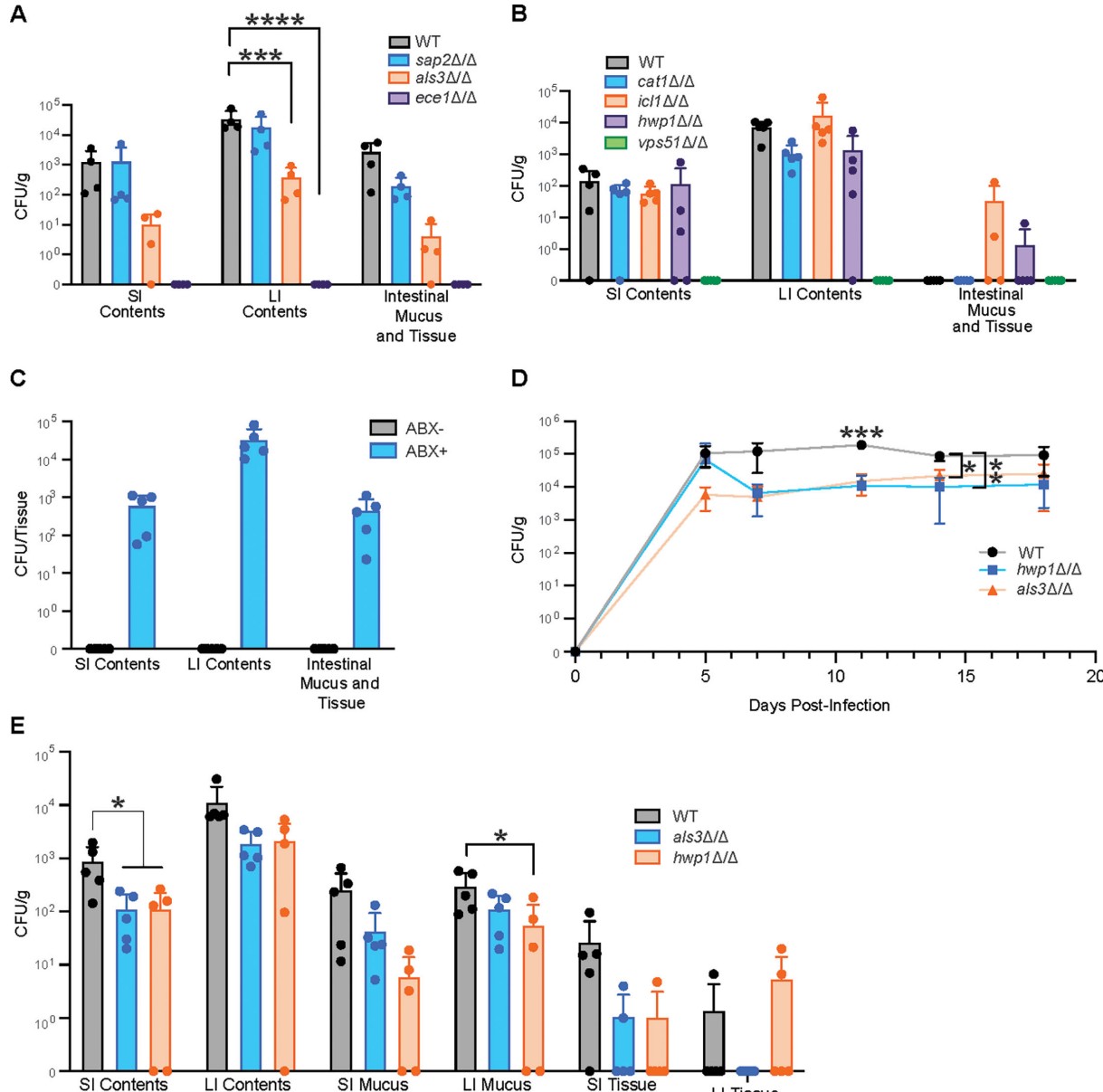

**FIG 6** Candidalysin, Als3, and Hwp1 contribute to robust colonization of the murine GI tract. (**A and B**) CFU/g *C. albicans* grown from the SI contents, LI contents, and homogenized intestinal mucus and tissue of mice infected with WT or virulence factor knockout strains of *C. albicans* (CLCA10). Samples were collected 14 days post-infection. (**C**) CFU *C. albicans* cultured from SI contents, LI contents, and intestinal mucus and tissue of mice infected with *ece1Δ/Δ C. albicans* (CLCA10) via oral gavage. Ampicillin and streptomycin were given to ABX + mice for 7 days prior to infection. Samples were collected 14 days after infection. (**D and E**) CFU/g *C. albicans* grown from (D) fecal pellets or (E) the SI and LI contents, SI and LI mucus, and homogenized SI and LI tissue of mice infected with WT, als3Δ/Δ, or hwp1Δ/Δ strains of *C. albicans* (CLCA10). Intestinal contents, mucus, and tissue were collected 21 days post-infection. Graphs show mean ± SD from 5 mice per group. Significance determined using (**A, B, and D**) one-way or (C) two-way ANOVAs with Dunnett's test. *$P < .05$, **$P < .01$, ***$P < .001$, ****$P < .0001$. Data representative of two experiments yielding comparable results.

that disrupt the microbial or immunologic state of the host or cause intestinal damage. We have shown that colonization of the GI tract by *C. albicans* is not host- or method-specific, as our long-term colonization model proved successful regardless of how the fungus was cultured *ex vivo*, and independent of host sex, strain, or commercial source. We further show that the ability to robustly colonize the murine GI tract under healthy conditions is not shared by all human-associated fungal species and strains, as other yeast species, including *S. cerevisiae*, *N. glabratus*, and other *Candida* spp., were unable to persist in the normal mouse GI tract in our experimental settings. In addition, the

hypha-associated peptide toxin candidalysin is required for robust colonization of and long-term persistence within the GI tract. Together, our findings indicate that the lumen and mucus layer of the intestine are major sites of long-term and possibly life-long colonization of the mouse, providing a mechanism for further understanding recrudescent infection and the remarkable commensal growth of this unique fungus in diverse human populations.

Studies of *C. albicans* colonization of the GI tract have generally used antibiotic treatment or germ-free conditions to enhance the fungal burden or extend *C. albicans* gut colonization. These methods are relevant for understanding the pathogenesis of invasive *Candida* spp. infections, since fungal expansion due to antibiotic use or immunosuppression greatly increases the risk for systemic infection (76–79). However, it is becoming increasingly clear that the presence of an intact microbiome alters *C. albicans* behavior in the GI tract as well as expression of the proteins necessary for colonization, leading to both synergistic and antagonistic interactions with gut bacteria (71, 80, 81). Understanding *C. albicans* dynamics and especially the transition between the commensal and pathogenic states requires models of *C. albicans* gut colonization with both intact and disrupted microbiomes. Accordingly, other groups have acknowledged this concept and are transitioning to other forms of colonization facilitation, most notably through dietary changes (71). However, our results show that *C. albicans* can effectively colonize mice consuming diets of variable composition, as we observed that *C. albicans* colonized mice consuming a variety of diets, including standard chow, as well as low- and high-fat purified diets.

Previous investigations by McDonough et al. identified strains of *C. albicans* that readily colonize the GI tract of mice without requiring disruption of the microbiome (26). Although the authors report that the *C. albicans* strain SC5314 was unable to colonize the gut in a C57BL/6J mouse cohort housed in Rhode Island due to its poor cathelicidin CRAMP (LL-37) resistance, they further show that SC5314 persists in the feces of C57BL/6J mice from a New York cohort out to 48 days. Comparing the microbiomes of the two cohorts, McDonough et al. show that the gut bacterial compositions differ substantially, with the NY cohort showing predominant Firmicute species, whereas in the RI cohort Bacteroidetes predominate (26). Comparing these results to our Houston cohort of C57BL/6J mice (Fig. 1E), the gut microbiome shows a slight predominance of Firmicute bacteria (Clostridiales and Lactobacillales) or at least an even distribution between Firmicutes and Bacteroidetes/Bacteroidales. This comparison thus suggests that *C. albicans* strain SC5314 is more likely to persist in the context of a gut microbiome that is substantially enriched in Firmicute species.

Although strain SC5314 was able to persist in the unmanipulated GI tract of our C57BL/6J cohort, fungal burdens were nonetheless reduced as compared to our clinical isolate strain CLCA10, further in keeping with the findings of McDonough et al. (26). As we found that the methods used to prepare the yeast inoculum did not affect colonization by SC5314 in our model, the subtle methodological differences between these studies are unlikely to contribute to the reduced colonization efficiency of SC5314. Interestingly, recent studies characterizing a variety of *C. albicans* strains derived from the human intestine have demonstrated vast phenotypic and genetic diversities of gut-resident *C. albicans* strains (6, 82, 83), showing that both "commensal" and "hypervirulent" strains are capable of colonizing the human intestine. Overall, these observations indicate that many strains of *C. albicans* are well-adapted to colonize both the human and mouse GI tracts, suggesting that the ability of *C. albicans* to colonize the mouse GI tract is determined less by the fungal strain used, but instead by distinct factors that likely include the precise composition of the (bacterial) microbiome.

Conversely, an additional potential factor influencing *C. albicans* colonization could be the presence of specific members of the gut mycobiome that outcompete *C. albicans* or otherwise influence its colonization potential. A recent study showed that the fungus *Kazachstania pintolopesii*, a common colonizer of the mouse GI tract, dominates the mycobiome upon colonization and imparts colonization resistance against multiple

other intestinal fungi, including *C. albicans* (84). Thus, in addition to genetic differences between mouse strains and the composition of the bacterial microbiome, mouse strain-dependent differences in *C. albicans* colonization could also reflect differences in the gut mycobiome.

In addition to *C. albicans*, many fungi, including other *Candida* species, inhabit the human intestinal microbiome (17, 23). However, among the fungal species and strains we examined, we found that colonization in the mouse GI tract in our model was not durable for non-*C. albicans* species. This may be due to differences in virulence factor expression between species, as we determined that candidalysin, Als3, and Hwp1 affected the degree to which *C. albicans* colonized the mouse GI tract. Als3 is uniquely expressed by *C. albicans* (85), and although Als3 deficiency was not sufficient to completely disrupt *C. albicans* colonization, *C. albicans* may use compensatory adhesins during intestinal colonization, which may also not be expressed by other *Candida* species. Furthermore, although orthologous ECE1 genes encoding peptide toxins like *C. albicans* candidalysin have been identified in other *Candida* species, including *C. tropicalis* (86), ECE1 gene expression is significantly lower than in *C. albicans*, suggesting potential differences in toxin production, biological function, and role during colonization and infection. Further investigation of the differences between these *Candida* species and their virulence factors is likely to reveal how *C. albicans* has adapted to colonize the GI tract. Furthermore, as we only tested one strain of each non-*C. albicans* species, additional strains should be evaluated for colonization fitness in the mouse GI tract to validate the colonization efficiency of these species.

A limitation of our study is that we did not investigate gut fungal morphology. In addition to the expression of select virulence factors, morphological changes are imperative for *C. albicans* persistence in the GI tract. Gut colonization was initiated in our model with yeast-phase cells in the inoculum, in part because previous studies have suggested that hyphal formation and expression of hypha-associated genes are detrimental for intestinal colonization by *C. albicans* in standard models, and that yeast-locked strains are hyper-fit in the intestine (30, 87–90). However, our observations that hypha-associated candidalysin is required for intestinal colonization and the fact that ECE1 is highly expressed during intestinal colonization of the murine GI tract (89) suggest that hypha formation is critical for effective intestinal colonization, depending on the model. In addition, our observation that the vps51Δ/Δ strain, which has been shown to be filamentation deficient (91), is unable to persist in the GI tract further supports the necessity of hyphal formation during intestinal colonization.

The discrepancy between our results and other publications is likely due to many of the previous studies using antibiotic-treated, neonatal, or germ-free mice, without intact, fully developed bacterial microbiomes. Indeed, in the study by Tso et al., the fitness advantage exhibited by non-filamenting *C. albicans* strains was negated by the introduction of an intact host microbiome (87). The need for the hyphal morphology and expression of ECE1 for colonization in a balanced microbiome is further supported by a more recent publication that showed that yeast-locked strains colonized well in antibiotic-treated hosts, but were unfit for colonization in the antibiotic-naïve host (71). While antibiotic treatment clearly enhances gut fungal burdens, it is unclear how such treatment influences *in situ* fungal morphology and the resulting effect on colonization potential. Nonetheless, these findings not only support the need for hyphal morphology and candidalysin production during colonization, but further emphasize the importance of microbiome inclusion when drawing conclusions about colonization of the GI tract by *C. albicans* under normal physiologic conditions.

These observations together suggest a model of *C. albicans* gut colonization in which fungal morphology is not monolithic, but in fact alternates between yeast and hyphal forms to both maintain colonization and disease potential across diverse gut conditions. This, in turn, suggests that the terms commensalism, colonization, and infection, although defined somewhat differently between investigators, should not be strictly linked to *C. albicans* morphology. In support of this, both fungal morphologies have been

found in colonization models with (88) and without (71) antibiotic treatments. In the latter study, hyphal formation, although known to be required for invasion, was shown to be a commensal factor. Moreover, colonization without symptoms may include low levels of invasion (92).

Our observations not only identify qualities of *C. albicans* that enhance its ability to grow as a commensal but also indicate a significant interaction between *C. albicans* and the intestinal mucus layer. Intestinal mucus is an essential component in the maintenance of healthy interactions between the host and the microbiome. Layers of gel-forming mucins protect the intestinal epithelium from pro-inflammatory exposure to the gut microbiota, while bacteria break down and utilize polysaccharides in the mucus (93). Previous studies have shown that gel-forming mucins inhibit, but do not abrogate, *C. albicans* virulence, most notably its ability to form hyphae (94, 95). However, our data demonstrate that live, culturable *C. albicans* cells are enriched in the mucus layer of both the small and large intestine, suggesting that *C. albicans* readily, and likely obligatorily, inhabits this niche to colonize mice.

While the virulence-attenuating properties of mucus may not prevent *C. albicans* from colonizing the mucus layer, they likely play a role in preventing *C. albicans* from interacting with the underlying epithelial cells via hyphal virulence factors for attachment and invasion (62, 63, 66). Indeed, we found that while *C. albicans* was enriched in the mucus of the intestine, it was almost entirely excluded from the intestinal tissue. However, the factors that establish this relationship and the benefits experienced by both *C. albicans* and the host because of mucus colonization are not completely understood. One of these factors may be the programming of host immune responses, as intestinal mucosal-associated fungi, such as *Candida* spp., have been shown to promote barrier-enhancing immune responses and to protect from infection by pathogenic bacteria in mice (96). Moreover, factors that disrupt either host immunity or the gut microbiome promote invasive disease or immune dysregulation due to *C. albicans* (97).

An additional limitation of our study is that all experiments were done in a murine model. Therefore, some of our observations may not be fully applicable to the mechanisms of commensal growth of *C. albicans* in the human intestine. Indeed, although we found that strains of the non-*C. albicans* species that we tested were unable to colonize the murine GI tract beyond 14 days, other *Candida* species have been isolated from human intestinal contents (23, 60). Whether the isolation of any *Candida* species from the human intestinal tract represents transient or long-term colonization requires further study.

There have been many attempts to design fungal vaccines to prevent or treat mucosal and systemic *C. albicans* infection, and although there are currently no licensed vaccines for fungal diseases, a vaccine candidate (NDV-3A) is pending phase III clinical trials following significant efficacy shown in phase II trials in the treatment of vulvovaginal candidiasis (98). Considering the poor activity of many antifungal drugs against *Candida* species and the health risks incurred by chronic antifungal administration (99, 100), this vaccine candidate represents a promising advancement in the fight against mucosal candidiasis. However, it remains unclear whether candidate vaccines affect intestinal colonization or dissemination from the gut, an important factor to consider as *C. albicans* in the gut is the most common reservoir for disseminated candidiasis (21–23). Amphotericin B administration was insufficient to clear *C. albicans* from the mouse intestine, although it is possible that more aggressive regimens might eventually be found that are sufficient to completely clear *C. albicans in vivo*. Nonetheless, antifungal treatment alone is likely to be costly, hazardous to the host, and rendered inconsequential due to re-infection. Therefore, it is imperative that we investigate further the mechanisms of human intestinal commensal growth of *C. albicans* to identify effective eradication strategies. Antifungals, vaccines, or antibodies targeting candidalysin (101) and adhesins such as Hwp1 and Als3 are a promising start, but additional critical colonization factors await discovery.

## MATERIALS AND METHODS

### Mice

Male and female C57BL/6, BALB/c, and 129S1/SvlmJ mice were purchased from the Baylor College of Medicine Center for Comparative Medicine, Jackson Laboratories, or Taconic Biosciences. A total of 278 mice were used in the reported experiments (273 females and 5 males). Mice were housed at the American Association of Laboratory Animal Care-accredited vivarium at Baylor College of Medicine under specific-pathogen-free conditions and used at 6–8 weeks of age. Mice were fed standard chow (LabDiet, St. Louis, MO) unless otherwise noted. Animal research protocols were approved by the Institutional Animal Care and Use Committee of Baylor College of Medicine and followed federal guidelines.

### Fungal strains

Experiments using wild-type *C. albicans* were conducted using a wild-type clinical isolate strain obtained from a patient with symptomatic asthma (CLCA10) confirmed to be *C. albicans* by whole-genome sequencing unless otherwise noted. Other wild-type strains of *C. albicans* used include SC5314, BWP17 + Clp30, CAI4 + Clp10 (provided by Dr. Bernhard Hube, Hans-Knoell Institute), and 529L (provided by Dr. Julian Naglik, King's College London). Virulence factor-deficient *C. albicans* strains (sap2△/△, als3△/△, ece1△/△, cat1△/△, icl1△/△, hwp1△/△, and vps51△/△) were generated using CLCA10 as the parental strain. Experiments involving non-*C. albicans* yeast strains were conducted using *C. tropicalis* ATCC 750, *N. glabratus* ATCC 2001, *C. auris* 0390 (provided by Dr. Gill Diamond, University of Florida), and *S. cerevisiae* EM93 (provided by Dr. Michael Lorenz, University of Texas Health Science Center at Houston).

### Construction of virulence factor-deficient strains

The clinical isolate of *C. albicans* CLCA10 was used for all gene deletions. A modified CRISPR-Cas9 approach (102) was used to delete both alleles of the gene of interest simultaneously. In short, CAS9 was amplified using PCR from pV1093. We selected specific guide RNAs and PCR-amplified them to be expressed under the control of the SNR52 promoter. The repair constructs containing the SAT1-FLP1 cassette were amplified using PCR from pSFS2 (103), using long primers with 50 bp-homologous flanking fragments specific to the target gene (primer sequences in Table S1). All PCR products were column-purified, and 500 ng of each fragment (CAS9, gene-specific sgRNA, gene-specific repair construct) was used to electroporate *C. albicans* (104). Transformants were chosen on YPD-Nat, and the correct gene deletion was confirmed by PCR. The accurate clones were made Nat-sensitive by overnight growth on YPM (2% maltose) to excise the SAT1-FLP1 cassette.

### Fungal cultivation

For all *in vivo* experiments, fungal cultures (including all *Candida* spp., *N. glabratus*, and *S. cerevisiae*) were grown by culturing 250,000 CFU yeast (stored in liquid nitrogen) in 50 mL YPD broth for 16 h at 37°C with shaking (120 rpm). The culture was then passed through a 40 µm mesh, centrifuged, and resuspended in PBS. After counting, the fungal suspension was diluted to $5 \times 10^7$ cells/mL with PBS. For conditions using stationary phase yeast, the *C. albicans* culture was grown for 24 h at 37°C with shaking (120 rpm). For conditions using plate-grown yeast, 250,000 CFU yeast were plated on Sabouraud's agar containing 50 µg/L chloramphenicol and cultured overnight at 37°C. Colonies were collected from the plate, resuspended in PBS, and processed as previously described.

For *in vitro* experiments using TR146 cells, *C. albicans* strains were cultured in YPD broth overnight at 30°C with shaking (180 rpm). The culture was then washed twice and resuspended in PBS, and absorbance was measured using a spectrophotometer (600 nm). Resuspended samples were adjusted to the desired cell density. A multiplicity

of infection (MOI) of 10 was used for 4 h experiments, and an MOI of 0.1 was used for 24 h experiments.

## Feeding of high-fat diet

Some mice were fed a purified diet *ad libitum* containing either 10% or 60% calories derived from fat (D12450J and D12492, respectively, Research Diets, New Brunswick, NJ) for 5 days prior to infection, and for the duration of the indicated experiment.

## Antifungal and antibacterial administrations

Select mice were given antifungal treatment after infection. Mice were administered 100 µg of liposome-encapsulated amphotericin B (AmBisome; Gilead Sciences, Foster City, CA) in PBS (or PBS only for vehicle-treated mice) daily via intraperitoneal injection. In addition, mice were given MediGel Sucralose cups (ClearH$_2$O, Westbrook, ME) containing 17.85 mg/mL AmBisome (or PBS only for vehicle-treated mice) to be consumed *ad libitum* and which were replaced daily.

In experiments where antibacterials were administered, mice were administered streptomycin (20 µg/mouse) in water via oral gavage 7 days prior to colonization. In addition, mice were given water with ampicillin (1 g/L) *ad libitum* for the 7 days prior to fungal colonization. Upon colonization, the mice were returned to their standard supply of normal chow and sterilized water.

## Murine gastrointestinal colonization and determination of fungal burden

For mouse infections, 6- to 8-week-old C57BL/6 female mice were used for all experiments with the exception of one experiment to rule out sex differences, where male mice were also used (Fig. 2A). Mice housed in specific pathogen-free conditions were given 5 × 10$^6$ *C. albicans* yeast cells (in 100 µL PBS) via oral gavage. PBS-treated mice were gavaged first, and cage changes for PBS-treated mice were conducted before fungal-infected mice to minimize the potential for cross-contamination. Fourteen days after infection (or on specified days during long-term experiments), the mice were euthanized and tissue samples were collected for determination of intestinal fungal burdens. The small intestine and large intestine (including cecum and colon) were collected, and lumens were flushed with 10 mL PBS, after which the liquid was centrifuged and the pellets frozen at −20°C. The flushed intestinal tissue was then minced and incubated in PBS containing 0.1 mg/mL proteinase K (Invitrogen, Waltham, MA) for 3 h at 37°C with shaking (100 rpm) after which the samples were collected and stored at −20°C. For experiments separating intestinal mucus from tissue, flushed intestinal tissue was first incubated in 1 mM DTT in PBS for 10 min at 37°C with shaking (200 rpm). The tissue samples were then vigorously shaken by hand, and the supernatant was collected and centrifuged to isolate a mucus pellet, which was frozen at −20°C (intestinal tissue was then digested with proteinase K as previously described). Samples were later thawed at room temperature, and intestinal tissue was homogenized and centrifuged. Intestinal contents, mucus, and intestinal tissue (pellet after centrifugation) were weighed, resuspended in 300 µL PBS, and plated on Sabouraud's agar containing 50 µg/L chloramphenicol. The next day, colony-forming units were counted and normalized by grams of sample.

## Histology

Formalin-fixed intestinal segments (collected three days post-infection) were sectioned, mounted on slides, and stained using hematoxylin and eosin. Slides were imaged using an EVOS M5000 Imaging System (Thermo Fisher Scientific, Waltham, MA).

## 16S sequencing of murine fecal pellets

Murine fecal pellets were collected immediately before infection (pre-colonization) and 14 days post-infection (post-colonization). Genomic DNA extraction and 16S rRNA sequencing of the V3 and V4 regions were performed on murine fecal pellets by GeneWiz (Azenta Life Sciences, South Plainfield, NJ). Sample QC was performed using a Nano-Drop 2000 Spectrophotometer (ThermoFisher Scientific, Waltham, MA), and samples were analyzed on an Illumina platform using the 2 × 250 bp paired-end configuration. Bioinformatic analysis was performed by GeneWiz using Cutadapt (v1.9.1), Vsearch (1.9.6), and Qiime (1.9.1) software.

## TR146 cell culture and infection

For *in vitro* virulence assays, TR146 human buccal epithelial squamous cell carcinoma cells (ECACC 10032305) were used (105). Cells were seeded at $5 \times 10^5$ cells/mL in Dulbecco's Modified Eagle Medium (Gibco, Waltham, MA) supplemented with 15% fetal bovine serum and 1% penicillin-streptomycin (Sigma Aldrich, St. Louis, MO) at 37℃ with 5% $CO_2$. After 24 h, when cells were 90%–100% confluent, cells were serum-starved overnight in fresh, serum-free medium and infected with WT *C. albicans* (SC5314, 529L, an original CLCA10 isolate, or a passaged stock of CLCA10) at the designated MOI or treated with PBS as a control.

## LDH release assay

TR146 cells were infected with *C. albicans* at an MOI of 0.1 for 24 h. PBS was used as the vehicle control, and 2% Triton-X 100 was used as a positive control. Exhausted cell culture media was collected and clarified by centrifugation ($350 \times g$, 10 min, 4℃). LDH activity was measured using a CytoTox 96 Non-Radioactive Cytotoxicity Assay (Promega, Madison, WI), and a standard curve was created using porcine lactate dehydrogenase (Sigma-Aldrich, St. Louis, MO). Data are representative of three independent experiments.

## RNA isolation and RT-qPCR

TR146 cells were infected with *C. albicans* at an MOI of 10 for 4 h. PBS was used as the vehicle control. After being washed with cold PBS, the cells were collected in PBS via scraping. Cell pellets were isolated via centrifugation ($10,000 \times g$, 1 min, 4℃) and flash-frozen in liquid nitrogen. After resuspension in 300 µL AE buffer, pellets were combined with 20 µL of 25% SDS and 300 µL of phenol (pH 4.3). Samples were incubated at 65℃ for 20 min with vortexing each minute before incubation on ice for 5 min. After centrifugation (13,000 rpm, 15 min, 4℃), the supernatant was collected, mixed via vortexing with 325 µL chloroform, and centrifuged (8,000 rpm, 10 min, 4℃). The supernatant was collected and vortexed with 400 µL of chloroform and centrifuged (8,000 rpm, 15 min, 4℃). The supernatant was then mixed with 20 µL sodium acetate (3 M) alongside 200 µL of 100% isopropanol. Tubes were gently inverted and incubated at −20℃ for 1 h. Samples were centrifuged (13,000 rpm, 20 min, 4℃), and pellets were washed with 70% ethanol and re-centrifuged (13,000 rpm, 20 min, 4℃). Pellets were air-dried and re-suspended in RNAse-free water, and RNA concentration was quantified using a NanoDrop 2000 Spectrophotometer (ThermoFisher Scientific, Waltham, MA). cDNA was synthesized using a High-Capacity cDNA Reverse Transcriptase kit (ThermoFisher Scientific, Waltham, MA). Real-time qPCR was performed using SYBR Green (Qiagen, Hilden, Germany) on a QuantStudio 7 Flex Real-Time PCR System, and gene expression was calculated using the threshold cycle ($\Delta\Delta C_T$) method, normalized to actin.

## Western blotting

TR146 cells were infected with *C. albicans* at an MOI of 10 for 4 h. PBS was used as the vehicle control. After incubation, cell culture plates were placed on ice, the culture medium was removed, and the cells were washed twice with cold PBS. RIPA lysis buffer supplemented with phosphatase and protease inhibitors (ThermoFisher Scientific, Waltham, MA) was added to cells for 2 min and adherent cells were scraped from the surface of the plate using a cell scraper and gently transferred into a pre-cooled microcentrifuge tube, which was incubated on ice for 30 min, followed by centrifugation (13,300 × *g*, 10 min, 4°C). Protein concentration in lysates was quantified using a Microplate Bicinchoninic Acid Protein Acid Kit (ThermoFisher Scientific, Waltham, MA). Cell lysates (10 µg) were resolved by electrophoresis onto SDS-PAGE 4-12% gradient gels (Invitrogen, Waltham, MA), transferred onto nitrocellulose membranes (Bio-Rad, Hercules, CA), and blocked for 1 h with gentle shaking. Membranes were incubated with the primary antibody with gentle agitation overnight at 4°C. The primary antibodies used were as follows: pEGFR (3777), Total EGFR (2232), Phospho-DUSP1/MKP1 (2857), and c-Fos (2250) from Cell Signaling Technology (Danvers, MA) and α-actin (MAB1501) from Merck Millipore (Burlington, MA). Membranes were subsequently probed with secondary antibody for 1 h at RT. The secondary antibodies used were as follows: Peroxidase-conjugated AffiniPure Goat anti-mouse (115-035-062) or anti-rabbit (111-035-003) from Jackson Immunoresearch (West Grove, PA). Proteins were detected with the Immobilon Western Chemiluminescent HRP Substrate (Merck Millipore, Burlington, MA) and developed with an Odyssey Fc Imaging System (LI-COR Biosciences, Lincoln, NE). Data are representative of three independent experiments.

## Statistical analysis

Data were analyzed using GraphPad Prism 10 and are presented as means ± standard deviation. Significant differences are presented as *P* values of < 0.05, as measured by Student's t-tests with Holm-Šídák correction, or one-way or two-way analysis of variance followed by Dunnett's test for multiple comparisons.

## ACKNOWLEDGMENTS

The content of this manuscript is solely the responsibility of the authors and does not necessarily represent the official views of the National Institutes of Health. Research reported herein was supported by US NIH grants HL140398, R01AI135803, 1K01AG083119, 1T32HL139425, and 5T32HL007747 and the National Institute of General Medical Sciences of the National Institutes of Health under award number T32GM136554. J.R.N. was supported by grants from the Wellcome Trust (214229_Z_18_Z) and National Institutes of Health (DE022550). Illustrative figures were generated using BioIcons.com, with changes made using Inkscape (1000_ml_Erlenmeyer_flask icon by Xavax (https://commons.wikimedia.org/wiki/User_talk:Xavax) is licensed under CC0 https://creativecommons.org/publicdomain/zero/1.0/; petri-dish-top-yellow icon by Servier (https://smart.servier.com/) is licensed under CC-BY 3.0 Unported https://creativecommons.org/licenses/by/3.0/; microtube-closed icon by DBCLS (https://togotv.dbcls.jp/en/pics.htmlhttps://togotv.dbcls.jp/en/pics.html) is licensed under CC-BY 4.0 Unported https://creativecommons.org/licenses/by/4.0/).

## AUTHOR AFFILIATIONS

[1]Department of Medicine, Baylor College of Medicine, Houston, Texas, USA

[2]Department of Microbiology and Molecular Genetics, McGovern Medical School, The University of Texas Health Science Center at Houston, Houston, Texas, USA

[3]Centre for Host-Microbiome Interactions, Faculty of Dentistry, Oral & Craniofacial Sciences, King's College London, London, United Kingdom

[4]Department of Microbial Pathogenicity Mechanisms, Leibniz Institute for Natural Product Research and Infection Biology, Hans Knoll Institute (HKI), Jena, Germany

[5]Cluster of Excellence Balance of the Microverse, Friedrich Schiller University Jena, Jena, Germany

[6]Institute of Microbiology, Friedrich Schiller University Jena, Jena, Germany

[7]Dan L. Duncan Comprehensive Cancer Center and Biology of Inflammation Center, Baylor College of Medicine, Houston, Texas, USA

[8]Center for Translational Research on Inflammatory Diseases, Michael E DeBakey VA Medical Center, Houston, Texas, USA

[9]Department of Pathology and Immunology, Baylor College of Medicine, Houston, Texas, USA

## AUTHOR ORCIDs

Kelsey E. Mauk http://orcid.org/0000-0001-5985-4148
Pedro Miramón http://orcid.org/0000-0002-6867-7697
Michael C. Lorenz http://orcid.org/0000-0002-7881-8027
Léa Lortal https://orcid.org/0009-0004-6221-7763
Julian R. Naglik http://orcid.org/0000-0002-8072-7917
Bernhard Hube http://orcid.org/0000-0002-6028-0425
Lynn Bimler http://orcid.org/0000-0002-7473-9698
David B. Corry http://orcid.org/0000-0001-9729-1016

## FUNDING

| Funder | Grant(s) | Author(s) |
| --- | --- | --- |
| National Institutes of Health | T32GM136554, HL140398, R01AI135803, K01AG083119, T32HL139425, T32HL007747 | Lynn Bimler |
| | | Farrah Kheradmand |
| | | David B. Corry |
| Wellcome Trust | 214229_Z_18_Z | Léa Lortal |
| | | Julian R. Naglik |
| National Institutes for Health and Care Research | DE022550 | Julian R. Naglik |

## AUTHOR CONTRIBUTIONS

Kelsey E. Mauk, Conceptualization, Data curation, Formal analysis, Investigation, Methodology, Validation, Visualization, Writing – original draft, Writing – review and editing | Pedro Miramón, Resources, Writing – review and editing | Michael C. Lorenz, Conceptualization, Data curation, Resources, Supervision, Writing – review and editing | Léa Lortal, Investigation, Methodology, Validation, Writing – review and editing | Julian R. Naglik, Investigation, Methodology, Supervision, Validation, Writing – review and editing | Bernhard Hube, Conceptualization, Supervision, Writing – review and editing | Lynn Bimler, Conceptualization, Data curation, Formal analysis, Investigation, Methodology, Project administration, Resources, Validation, Visualization, Writing – original draft, Writing – review and editing | Farrah Kheradmand, Conceptualization, Formal analysis, Writing – review and editing | David B. Corry, Conceptualization, Formal analysis, Funding acquisition, Project administration, Resources, Supervision, Writing – original draft, Writing – review and editing

## DATA AVAILABILITY

Fungal strains generated as part of this project will be available to qualified researchers upon request to Dr. David B. Corry. Raw reads from 16S sequencing are available on the NCBI Sequence Read Archive under BioProject accession number PRJNA1197521.

## ADDITIONAL FILES

The following material is available online.

### Supplemental Material

**Supplemental material (Spectrum00567-25-S0001.pdf).** Fig. S1 to S4; Table S1.

### Open Peer Review

**PEER REVIEW HISTORY (review-history.pdf).** An accounting of the reviewer comments and feedback.

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
