## [Reviewer comments · Microbiology Spectrum]

Microbiology Spectrum

Commensal Colonization of *Candida albicans* in the Mouse Gastrointestinal Tract is Mediated Via Expression of Candidalysin and Adhesins

Kelsey Mauk, Pedro Miramón, Michael Lorenz, Léa Lortal, Julian Naglik, Bernhard Hube, Lynn Bimler, Farrah Kheradmand, and David Corry

Corresponding Author(s): David Corry, Baylor College of Medicine

Review Timeline:

Submission Date:	March 21, 2025
Editorial Decision:	April 29, 2025
Revision Received:	May 13, 2025
Editorial Decision:	June 9, 2025
Revision Received:	June 19, 2025
Accepted:	June 24, 2025

Editor: Dhammika Navarathna

Reviewer(s): The reviewers have opted to remain anonymous.

Transaction Report:

DOI: <https://doi.org/10.1128/spectrum.00567-25>

Re: Spectrum00567-25 (Commensal Colonization of *Candida albicans* in the Mouse Gastrointestinal Tract is Mediated Via Expression of Candidalysin and Adhesins)

Dear Dr. David Corry:

Thank you for the privilege of reviewing your work. Below you will find my comments, instructions from the Spectrum editorial office, and the reviewer comments.

Authors need to put more effort to address original reviewers' comments. Specifically, to properly acknowledge previous studies that have rigorously shown that *C. albicans* can colonize a mouse without microbiota perturbation. In addition, all previous reviewers requested a discussion on why the SC5314 strain seems to stably colonize in their mouse models, when several previous studies found it was a poor colonizer in multiple mouse backgrounds and in multiple mouse facilities. Authors need to make substantive changes to the text in this revised document, and instead directly emphasizing that these findings as if it is the first to directly test this question. Please revisit finding that non-*albicans* were not able to colonize SPF mice in your model and cite previous reports as well.

Revision Guidelines

Sincerely,
Dharmika Navarathna
Editor
Microbiology Spectrum

Reviewer #1 (Comments for the Author):

See the attached.

Reviewer #2 (Comments for the Author):

This study investigates fungal, dietary, and strain factors that regulate the persistence and colonization niche of different *C. albicans* strains in the murine GI tract. The study reinforces previous work from multiple labs showing that *C. albicans* can indeed persist in the mouse gut without antibiotics. Strengths of this study include the detailed investigation of the colonization location of *C. albicans* within the gut (SI, LI lumen, mucus, tissue-associated), and demonstration that preparation of the *C. albicans* inoculum has only a minor impact on the colonization levels. However, there are major concerns for how the results of this study are discussed, what the data does and does not support, and how these findings support or refute previous investigations of *C. albicans*-GI colonization studies. The points and statements that need to be qualified, changed, or removed are described below:

The authors should be cautious in referring to CLCA10 as "commensal-like" throughout the paper, as this is a tricky designation to make based only on phenotypic evidence that a strain is less damaging/virulent. Indeed, Li et al 2022 and Anderson et al 2023 both found that strains isolated from the mouth or gut of healthy individuals have a range of pathogenic potentials, with many being virulent in animal models of infection and damaging in co-culture assays with macrophages. Furthermore, the CLCA10 was isolated from a patient and niche (asthmatic lung) that is likely atypical for a "commensal" strain.

The authors did not adequately address reviewer comments requesting a more detailed discussion of why SC5314 stably colonized mice in this study while it was found to be a poor colonizer in other studies. In particular, a nuanced comparison between these results and SC5314 colonization in McDonough et al 2021 is needed, as this study provides a comprehensive comparison of SC5314 colonization dynamics in different mouse backgrounds and different animal facilities. Discussion for why this study, microbiota/strain/colonization strategy, may have resulted in differing results is needed in the discussion.

The impact of SC5314 on the mouse intestinal bacterial community has been investigated in previous studies. This includes McDonough 2021, where they found no significant impact of this and other *C. albicans* strains on the bacterial community. This should be clearly highlighted in any discussion of Figure 1. Lines 164-166 currently claim that SC5314 has been shown to alter bacteria community diversity, citing a paper (Bertolini et al 2019). However, this cited study primarily focuses on the oropharyngeal microbiota in the setting of chemotherapy and has limited relevance to Figure 1. Instead, this section should note that these findings are consistent with previous studies showing minimal impacts of multiple *C. albicans* strains on the bacterial microbiota during homeostasis.

The data investigating other fungal species is not sufficient to claim that persistence in an SPF animal is "unique" or "specific" to *C. albicans*. Several fungal species, including *Kazachstania*, *C. tropicalis* and others, have been identified in the endogenous microbiota of mice in various facilities and in wild mouse populations (Iliev, Funari et al. 2012, Wang, Fan et al. 2018, Yeung, Chen et al. 2019, Jain, Ver Heul et al. 2021, Liao, Gao et al. 2024, Sekeresova Kralova, Donic et al. 2024). These fungi are viable and impact a wide range of health and disease phenotypes. It is inaccurate to make any sweeping claims about the ability of a fungus to colonize a complex microbiota based on a single colonization strategy, with a single strain, and a single mouse background. Any claim that persistence within an SPF animal is specific or unique to *C. albicans* should be removed.

The manuscript by Mauk et al. explores gut colonization patterns of a recently isolated clinical strain of *Candida albicans*, CLCA10. While the findings are specific to the strain and mouse model used, a key limitation is the generalization of the results, especially in light of the contrasting data reported in the literature for the commonly studied wild-type strain SC5314. Nevertheless, this study offers fresh insights that advance our understanding of strain-dependent complexity of *C. albicans* pathobiology.

It is not clear whether CLCA10 clinical isolate was obtained from a patient with symptoms of candidiasis or not. On what basis do the authors claim that CLCA10 prevents colonization as a commensal organism? For example, in lines 436–437: The authors claim that mucin inhibits hyphal formation, but does this also imply that yeast colonization is inhibited?

The authors did not specifically mention the hyphal phenotype (yeast or hyphae) observed in their histology samples. If mucin inhibits only hyphal colonization but not yeast forms, the statement in lines 437–440 should be revised accordingly.

Some data are presented for both CLCA10 and SC5314, while other data are provided for only one strain. There are several instances where the manuscript may confuse readers due to the lack of specification regarding which strain the results pertain to. The strain should be clearly indicated in the abstract, introduction (lines 109, 118, and 119), methods, and in lines 242–259, as appropriate.

Based on the observed data, an interesting hypothesis to consider is whether changes in colonization potential are influenced by fungal burden. Do the authors have any data to support or refute this hypothesis?

Abstract, lines 39–41: Due to the use of a clinical isolate, the conclusions may not be applicable to all wild-type strains. Therefore, the phrase 'This study concludes...' in lines 39–41 should be included to specify this limitation and reduce overgeneralization.

The authors claim that hyphae-associated ECE1 is highly expressed during hyphal colonization. However, is this true for all *Candida* species? The data in Supplementary Figure 1, particularly comparing SC5314 and CLCA10, do not clearly support this. Explain this discrepancy.

Line 193: How do you justify CLCA10 is a strain found in 'general community'?

In line 198: The authors claim that both sexes of mice were used; however, the methods section (lines 546–547) only mentions female mice. Could the authors clarify this discrepancy and include the number of animals used for each sex?

Line 316–317: Note that the *C. albicans* VPS51 knockout is a filamentation-deficient mutant (Hossain et al., 2022). Emphasizing and comparing this in your explanation for this and rest of the mutant strains used for this study could strengthen your findings. Comparing the degree of

filamentation across different phenotypes may reveal interesting correlations with colonization potential.

Thursday, May 1, 2025

Dr. Dhammika Navarathna
Microbiology Spectrum

Re: Revised manuscript, Spectrum00567-25

Dear Dr. Navarathna,

We thank the reviewers for their constructive comments and feedback on our manuscript. The following point-by-point response addresses the major concerns of the reviewers.

Reviewer #1:

1. *It is not clear whether CLCA10 clinical isolate was obtained from a patient with symptoms of candidiasis or not.*

Response: The Fungal Strains section of Methods has been expanded (new lines 519-521) to indicate that CLCA10 has been obtained from a symptomatic asthma patient. The term “candidiasis” can be ambiguous and has different meanings, although the term nearly universally implies disease. Given this, as *C. albicans* CLCA10 was isolated from a patient with symptomatic asthma, a common context for the recovery of *C. albicans* from human respiratory tract secretions^{1, 2, 3, 4, 5, 6, 7}, we believe “candidiasis” is appropriately applied to asthma.

2. *On what basis do the authors claim that CLCA10 prevents colonization as a commensal organism? For example, in lines 436–437: The authors claim that mucin inhibits hyphal formation, but does this also imply that yeast colonization is inhibited?*

Response: We did not assess the morphology of *C. albicans* during gut colonization, thus we cannot draw conclusions regarding the fungal morphology during colonization. Nonetheless, we document that *C. albicans*, regardless of strain, persists in the mouse gut lumen for up to 58 days without causing obvious gut pathology or systemic disease, meeting most definitions of “colonization” or “commensalism”. We do not state or imply that CLCA10 prevents colonization as a commensal organism; indeed, we are not entirely sure what the reviewer means by this statement. While it is true that mucins inhibit hyphal formation, this does not prevent colonization as again documented by our data showing persistent gut colonization by multiple *C. albicans* strains despite constant mucin production by the gut.

3. *The authors did not specifically mention the hyphal phenotype (yeast or hyphae) observed in their histology samples. If mucin inhibits only hyphal colonization but not yeast forms, the statement in lines 437–440 should be revised accordingly.*

Response: As mentioned, assessment of fungal morphology via histology was not conducted, as the methods used do not allow for accurate identification of fungi, and this was not the intended goal of the histologic analysis. While we found that *C. albicans* was enriched in the mucus layer via culturing, and the referenced studies^{8, 9} indicate that gel-forming mucins inhibit the formation of hyphae, we cannot conclude that fungi inhabiting the intestinal mucus layer of mice do so as yeasts, as we did not test this hypothesis directly. We have modified the discussion as follows: “Previous studies have shown that gel-forming mucins inhibit, but do not abrogate, *C. albicans* virulence, most notably its ability to form hyphae^{8, 9}. However, our data demonstrate that live,

culturable *C. albicans* cells are enriched in the mucus layer of both the small and large intestine, suggesting that *C. albicans* readily, and likely obligatorily, inhabits this niche to colonize mice” (lines 468-472 of revised manuscript)

4. *Some data are presented for both CLCA10 and SC5314, while other data are provided for only one strain. There are several instances where the manuscript may confuse readers due to the lack of specification regarding which strain the results pertain to. The strain should be clearly indicated in the abstract, introduction (lines 109, 118, and 119), methods, and in lines 242–259, as appropriate.*

Response: We thank the reviewer for this suggestion and have clarified the strain used in the noted sections and elsewhere.

5. *Based on the observed data, an interesting hypothesis to consider is whether changes in colonization potential are influenced by fungal burden. Do the authors have any data to support or refute this hypothesis?*

Response: We thank the reviewer for this indeed interesting hypothesis, but have not addressed it directly and cannot comment on it in this report.

6. *Abstract, lines 39–41: Due to the use of a clinical isolate, the conclusions may not be applicable to all wild-type strains. Therefore, the phrase 'This study concludes...' in lines 39–41 should be included to specify this limitation and reduce overgeneralization.*

Response: The phrase “This study concludes...” has been added to the specified statement.

7. *The authors claim that hyphae-associated ECE1 is highly expressed during hyphal colonization. However, is this true for all Candida species? The data in Supplementary Figure 1, particularly comparing SC5314 and CLCA10, do not clearly support this. Explain this discrepancy.*

Response: *ECE1* is highly expressed in hyphae produced by *C. albicans*, but not other *Candida* species. However, considering the context of this question, we assume that the reviewer may mean *C. albicans* strains. *ECE1* is highly expressed by SC5314 in the GI tract (see reference 91 in the revised manuscript). Moreover, *ECE1* is expressed in all *C. albicans* strains if they produce hyphae. In our hands (BH), all clinical and even environmental strains produce hyphae and candidalysin. Even poor in vitro hyphae producers, including in vitro-hyphae deficient mutants, produce hyphae in vivo as shown in the literature. Considering that hyphae are also beneficial for commensal growth, this is not surprising.

8. *Line 193: How do you justify CLCA10 is a strain found in 'general community'?*

Response: This strain was isolated from a sputum sample of an asthmatic individual receiving outpatient care, making it a human-isolated strain from a patient from the general community.

9. *In line 198: The authors claim that both sexes of mice were used; however, the methods section (lines 546–547) only mentions female mice. Could the authors clarify this discrepancy and include the number of animals used for each sex?*

Response: The methods section has been edited to state that male mice were only used in a single experiment, Figure 2A, and both the figure itself and the legend clearly indicate the number of mice used (5).

10. Line 316-317: Note that the *C. albicans* VPS51 knockout is a filamentation-deficient mutant (Hossain et al., 2022). Emphasizing and comparing this in your explanation for this and rest of the mutant strains used for this study could strengthen your findings. Comparing the degree of filamentation across different phenotypes may reveal interesting correlations with colonization potential.

Response: We thank the reviewer for bringing this reference to our attention, and for the insightful suggestion. We have added and expanded on this reference and a discussion on the role of filamentation in the Discussion section

Reviewer #2 (Comments for the Author):

1. *The authors should be cautious in referring to CLCA10 as "commensal-like" throughout the paper, as this is a tricky designation to make based only on phenotypic evidence that a strain is less damaging/virulent. Indeed, Li et al 2022 and Anderson et al 2023 both found that strains isolated from the mouth or gut of healthy individuals have a range of pathogenic potentials, with many being virulent in animal models of infection and damaging in co-culture assays with macrophages. Furthermore, the CLCA10 was isolated from a patient and niche (asthmatic lung) that is likely atypical for a "commensal" strain.*

Response: We acknowledge that the term “commensal-like” does not necessarily describe the full pathogenic potential of our *C. albicans* strain, as CLCA10, and other “commensal” strains, are fully capable of causing disease. In support of this point, SC5314, which has enhanced pathogenic potential, is also capable as existing as a commensal in our model of intestinal colonization. Our use of this term is not to separate “pathogenic” from “non-pathogenic” strains, but to describe the basic virulence phenotypes in comparison to other strains (SC5314 and 529L) to emphasize that diverse strains can be both pathogenic or commensal depending on the context.

2. *The authors did not adequately address reviewer comments requesting a more detailed discussion of why SC5314 stably colonized mice in this study while it was found to be a poor colonizer in other studies. In particular, a nuanced comparison between these results and SC5314 colonization in McDonough et al 2021 is needed, as this study provides a comprehensive comparison of SC5314 colonization dynamics in different mouse backgrounds and different animal facilities. Discussion for why this study, microbiota/strain/colonization strategy, may have resulted in differing results is needed in the discussion.*

Response: We thank the reviewer for allowing us to make a more detailed comparison to the study of McDonough, et al¹⁰. They show that *C. albicans* strain SC5314 persists in the feces of C57BL/6J mice from a New York cohort out to 48 days (Fig. 1B), but that the strain is lost from the feces of a distinct cohort of the same mouse strain housed in a Rhode Island facility by about day 23 (Fig. 1A). Comparing the microbiomes of the two cohorts, McDonough, et al. show that the gut bacterial compositions differ substantially, with the NY cohort showing predominant Firmicute species, whereas in the RI cohort Bacteroidetes predominate (Fig. 2C and B, respectively). Comparing these results to our Houston cohort of C57BL/6J mice (Fig. 1E of the current manuscript), the gut microbiome shows a slight predominance of Firmicute bacteria (Clostridiales and Lactobacillales) or at least an even distribution between Firmicutes and

Bacteroidetes/Bacteroidales. This comparison suggests, but does not prove, that *C. albicans* strain SC5314 is more likely to persist in the context of a gut microbiome that is substantially enriched in Firmicute species. We have added this comparison to the Discussion.

3. *The impact of SC5314 on the mouse intestinal bacterial community has been investigated in previous studies. This includes McDonough 2021, where they found no significant impact of this and other C. albicans strains on the bacterial community. This should be clearly highlighted in any discussion of Figure 1. Lines 164-166 currently claim that SC5314 has been shown to alter bacteria community diversity, citing a paper (Bertolini et al 2019). However, this cited study primarily focuses on the oropharyngeal microbiota in the setting of chemotherapy and has limited relevance to Figure 1. Instead, this section should note that these findings are consistent with previous studies showing minimal impacts of multiple C. albicans strains on the bacterial microbiota during homeostasis.*

Response: The results section discussing our investigations of the bacterial microbiota have been edited to include references to McDonough, 2021 and have clarified that our results using the CLCA10 strain are consistent with previous results from experiments using the SC5314 strain.

4. *The data investigating other fungal species is not sufficient to claim that persistence in an SPF animal is "unique" or "specific" to C. albicans. Several fungal species, including Kazachstania, C. tropicalis and others, have been identified in the endogenous microbiota of mice in various facilities and in wild mouse populations (Iliev, Funari et al. 2012, Wang, Fan et al. 2018, Yeung, Chen et al. 2019, Jain, Ver Heul et al. 2021, Liao, Gao et al. 2024, Sekeresova Kralova, Donic et al. 2024). These fungi are viable and impact a wide range of health and disease phenotypes. It is inaccurate to make any sweeping claims about the ability of a fungus to colonize a complex microbiota based on a single colonization strategy, with a single strain, and a single mouse background. Any claim that persistence within an SPF animal is specific or unique to C. albicans should be removed.*

Response: We acknowledge that our analysis of the colonization potential of non-*C. albicans* species is limited, particularly by the number of strains used, and that these strains may colonize mice in other contexts. We have adjusted our conclusions to better specify that the strains we used of these species were unable to colonize our model, rather than stating that these species are incapable of colonizing mice in general.

We again thank the reviewers for their insightful comments and suggestions and hope that the resulting changes to the manuscript are now sufficient to warrant publication.

Sincerely,

David B. Corry, MD (for the authors)

References:

1. **Mak, G., Porter, P.C., Bandi, V., Kheradmand, F. & Corry, D.B. Tracheobronchial mycosis in a retrospective case-series study of five status asthmaticus patients. *Clinical Immunology* 146, 77-83 (2013).**
2. **Porter, P.C. *et al.* Airway surface mycosis in chronic TH2-associated airway disease. *J Allergy Clin Immunol* 134, 325-331 (2014).**
3. **Takabatake, N., Seino, S., Nakamura, H. & Tomoike, H. A patient with allergic bronchopulmonary candidiasis showing a high serum level of soluble interleukin 2 receptors. *Japanese Journal of Thoracic Diseases* 35, 1271-1277 (1997).**
4. **Inoue, K. *et al.* A case of allergic bronchopulmonary candidiasis treated with amphotericin B inhalation. *Japanese Journal of Thoracic Diseases* 30, 352-357 (1992).**
5. **Hosen, H. Chronic asthma and rhinitis due to *Candida albicans*. *Ann. Allergy* 60, 272-273 (1988).**
6. **Gumowski, P., Lech, B., Chaves, I. & Girard, J.P. Chronic asthma and rhinitis due to *Candida albicans*, epidermophyton, and trichophyton. *Ann. Allergy* 59, 48-51 (1987).**
7. **Akiyama, K., Mathison, D.A., Riker, J.B., Greenberger, P.A. & Patterson, R. Allergic bronchopulmonary candidiasis. *Chest* 85, 699-701 (1984).**
8. **Kavanaugh, N.L., Zhang, A.Q., Nobile, C.J., Johnson, A.D. & Ribbeck, K. Mucins suppress virulence traits of *Candida albicans*. *mBio* 5, e01911 (2014).**
9. **Takagi, J. *et al.* Mucin O-glycans are natural inhibitors of *Candida albicans* pathogenicity. *Nat Chem Biol* 18, 762-773 (2022).**
10. **McDonough, L.D. *et al.* *Candida albicans* Isolates 529L and CHN1 Exhibit Stable Colonization of the Murine Gastrointestinal Tract. *mBio*, e0287821 (2021).**

Re: Spectrum00567-25R1 (Commensal Colonization of *Candida albicans* in the Mouse Gastrointestinal Tract is Mediated Via Expression of Candidalysin and Adhesins)

Dear Dr. David Corry:

Thank you for the privilege of reviewing your work. Below you will find my comments, instructions from the Spectrum editorial office, and the reviewer comments.

Revision Guidelines

Sincerely,
Dharmika Navarathna
Editor
Microbiology Spectrum

Reviewer #1 (Comments for the Author):

See the attached.

Reviewer #2 (Comments for the Author):

All of my concerns have been addressed.

The revised submission reflects progress, though a few points still require further clarification or refinement on the manuscript itself. Some points were addressed only in the 'Response to Reviewer Comments' but were not adequately addressed in the revised manuscript. Please ensure that these justifications and clarifications are appropriately incorporated in the manuscript text as pointed out below.

It is well known that the versatility of *Candida albicans* makes it difficult to distinguish its commensal and pathogenic behaviors across hosts [Infect. Immun. 84, no. 10 (2016): 2724-2739]. However, although the title suggests an exploration of 'commensal colonization,' a major caveat of the manuscript is the interchangeable use of the term 'colonization,' which obscures the critical distinction between colonization and invasion—and consequently, between commensalism and infection." This ambiguity is further compounded by the lack of phenotypic evidence—such as yeast or hyphal morphology—due to the absence of GMS staining in the histological specimens, limiting the ability to clearly differentiate these processes.

1. The authors stated, '*We did not assess the morphology of C. albicans during gut colonization, thus we cannot draw conclusions regarding the fungal morphology during colonization*'. The definition of 'commensal colonization' appears to be overlooked in the context of their chosen study model. The strain is a clinical isolate from a symptomatic patient while it is not clear why an invasive strain was justified as a commensal strain or from the 'general community (R#1: Comment 8 and R#2: Comment 1). Therefore, the authors should justify the use of a strain with a pathogenic potential to study 'commensal colonization' in a mice model. If the pathogenic potential '*depend on the context*', the authors should discuss the evidence for the commensal behavior of CLCA10 in their study including factors like host differences and inoculum size/fungal burden.
2. The limitations and speculations of the study pointed out previously by the reviewers should be addressed on the manuscript itself (for the reader) to avoid over generalization which include effect on the fungal burden (R#1: Comment 5) and lack of phenotypic evidence – yeast vs hyphal (R#1: Comment 3).
3. Line 418 (Marked up manuscript): correct the typo? "gut microbiome"?
4. Provide a comprehensive description of the research design (as per reporting guidelines for ASM journals), including but not limited to the number of mice used, along with their sex distribution, in the Methods section in addition to the figure legends. A detailed methodology is essential to ensure scientific rigor and reproducibility. Refer to the reporting guidelines for ASM journals.

Wednesday, June 18, 2025

Dharmika Navarathna
Editor
Microbiology Spectrum

Re: Spectrum00567-25R1

Dear Dr. Navarathna,

Thank you for the opportunity to respond to the ongoing concerns of Reviewer #1, which we address with the following point-by-point response:

1. *"It is well known that the versatility of Candida albicans makes it difficult to distinguish its commensal and pathogenic behaviors across hosts [Infect. Immun. 84, no. 10 (2016): 2724-2739]. However, although the title suggests an exploration of 'commensal colonization,' a major caveat of the manuscript is the interchangeable use of the term 'colonization,' which obscures the critical distinction between colonization and invasion—and consequently, between commensalism and infection." This ambiguity is further compounded by the lack of phenotypic evidence—such as yeast or hyphal morphology—due to the absence of GMS staining in the histological specimens, limiting the ability to clearly differentiate these processes. The authors stated, 'We did not assess the m_o_r_p_h_o_l_o_g_y_o_f_C._._a_l_b_i_c_a_n_s_d_u_r_i_n_g_g_u_t_c_o_l_o_n_i_z_a_t_i_o_n, t_h_u_s_w_e_c_a_n_n_o_t_d_r_a_w_c_o_n_c_l_u_s_i_o_n_s_r_e_g_a_r_d_i_n_g_t_h_e_f_u_n_g_a_l_m_o_r_p_h_o_l_o_g_y_d_u_r_i_n_g_c_o_l_o_n_i_z_a_t_i_o_n.' The definition of 'commensal colonization' appears to be overlooked in the context of their chosen study model."*

Response: We agree that the terms “commensalism”, “colonization” and “infection” need to be used carefully. We would argue that some of the points discussed by the reviewer and us are in fact a matter of opinion and not so much strict definitions; regardless, this is certainly a hot topic in the community (also discussed in a recent review by co-authors of our study (Schille et al. Nat Rev Microbiol. 2025)). Of note, almost all experimental approaches to study *C. albicans* colonization are based on mouse models. These models have been used to study commensalism (discussed in, for example: Mishra and Koh, Curr Opin Microbiol 2021; Kumamoto et al. Curr Opin Microbiol. 2020; Romo and Kumamoto J Fungi 2020; Schille et al. Nat Rev Microbiol. 2025).

As discussed in the study cited by this reviewer (Jabra-Rizk et al. Infect Immun 2016), the term “colonization” is used as follows: “As part of the commensal human microbiota, *C. albicans* asymptotically colonizes the gastrointestinal tract, oral cavity, and reproductive tract of healthy individuals.” This is exactly the case in our model, where we observe *C. albicans* persistence without obvious disease or inflammation in the mouse gut lumen for up to 58 days, meeting the above cited definitions of “asymptomatic colonization” and “commensalism”. We would further argue that the morphology of *C. albicans*, yeast or hyphae, is not relevant in this context as both morphologies are found in colonization models with (Witchley et al. Cell Host Microbe 2019) and without antibiotic treatments (Liang et al. Nature. 2024). In the latter, hyphal formation, although known to be required for invasion, is shown to be a commensal factor. It should further be noted that even colonization without symptoms may include low levels of invasion (Vautier et al., Cell Microbiol 2015: “Our data reveal that GI tract colonization favours the yeast form of *C. albicans*, that there is constitutive low

level systemic dissemination in colonized mice that occurs irrespective of fungal morphology”). Underscoring the findings of Vautier, *et al.*, we will soon be submitting a second manuscript where we demonstrate that the same gut colonization model with the same strain of *C. albicans* (CLCA10) leads to continuous dissemination of *C. albicans* to distant organs that, in contrast to the gut, provokes highly significant inflammation with profound physiological effects.

Essential elements of this response are now included in the Discussion (see especially lines 455-470).

2. *“The strain is a clinical isolate from a symptomatic patient while it is not clear why an invasive strain was justified as a commensal strain or from the ‘general community’ (R#1: Comment 8 and R#2: Comment 1). Therefore, the authors should justify the use of a strain with a pathogenic potential to study ‘commensal colonization’ in a mice model. If the pathogenic potential ‘d e p e n d o n t h e c o n t e x t’, the authors should discuss the evidence for the commensal behavior of CLCA10 in their study including factors like host differences and inoculum size/fungal burden.”*

Response: In addition to the related comments above, we would add here that the majority, if not all *C. albicans* strains have the potential to cause disease and that “invasive strains” isolated from the bloodstream in the clinics (like SC5314) have been previous gut colonization commensals as almost all infections are endogenous infections from endogenous colonizing strains. Pioneering work of Koh et al. PLOS Pathogens 2008 has shown that both neutropenia and GI mucosal damage are critical for allowing widespread invasive *C. albicans* disease. Furthermore, this model required “.... depleting resident GI intestinal flora with antibiotic treatment and achieving stable GI colonization levels of *C. albicans*...”. However, in our colonization model, we did not use antibiotics (or any further treatment) to demonstrate gut colonization of CLCA10 without any pathology or signs/symptoms of disease.

3. *“The limitations and speculations of the study pointed out previously by the reviewers should be addressed on the manuscript itself (for the reader) to avoid over generalization which include effect on the fungal burden (R#1: Comment 5) and lack of phenotypic evidence –yeast vs hyphal (R#1: Comment 3).”*

Response: We extensively addressed these concerns in the prior review and struggled to add additional commentary that might satisfy what the reviewer is asking for in the current revision. Please note the following additions to the Discussion:

-Line 428 A limitation of our study is that we did not investigate gut fungal morphology.

-line 430-1: Gut colonization was initiated in our model with yeast-phase cells in the inoculum, in part because previous studies have suggested that hyphal formation and expression of hypha-associated genes are detrimental....

-Line 449-51: While antibiotic treatment clearly enhances gut fungal burdens, it is unclear how such treatment influences in situ fungal morphology and the resulting effect on colonization potential. Nonetheless, these findings...

-Lines 455-70: These observations together suggest a model of *C. albicans* gut colonization in which fungal morphology is not monolithic, but in fact alternates between yeast and hyphal forms to both maintain colonization and disease potential across diverse gut conditions. In fact, both fungal morphologies have been found in colonization models with⁹⁴ and without antibiotic treatments⁷³. In the latter study, hyphal formation, although known to be required for invasion, was shown to be a

commensal factor. It should further be noted that even colonization without symptoms may include low levels of invasion⁹⁵.

4. *“Line 418 (Marked up manuscript): correct the typo? “gut microbiome”?”*

Response: Corrected, thank you for picking up this typo.

5. *“Provide a comprehensive description of the research design (as per reporting guidelines for ASM journals), including but not limited to the number of mice used, along with their sex distribution, in the Methods section in addition to the figure legends. A detailed methodology is essential to ensure scientific rigor and reproducibility. Refer to the reporting guidelines for ASM journals.”*

Response: Methods have been updated to include the specific number of mice use and their sex (Line 500: A total of 278 mice were used in the reported experiments (273 female, 5 male)); and more specific wording as to precautions against cross-contamination (Lines 570-1: PBS-treated mice were gavaged first and cage changes for PBS-treated mice were conducted before fungal-infected mice to minimize the potential for cross-contamination.).

Thank you again for the opportunity to respond to the referee concerns, which we hope are now fully addressed.

Sincerely,

David B. Corry, MD

Re: Spectrum00567-25R2 (Commensal Colonization of *Candida albicans* in the Mouse Gastrointestinal Tract is Mediated Via Expression of Candidalysin and Adhesins)

Dear Dr. David Corry:

Authors have addressed concerns raised after R1 revision.

Your manuscript has been accepted, and I am forwarding it to the ASM production staff for publication. Your paper will first be checked to make sure all elements meet the technical requirements. ASM staff will contact you if anything needs to be revised before copyediting and production can begin. Otherwise, you will be notified when your proofs are ready to be viewed.

Sincerely,
Dharmika Navarathna
Editor
Microbiology Spectrum